# Genetics of circulating proteins in newborn babies at high risk of type 1 diabetes

Mauro Tutino[1,18], Nancy Yiu-Lin Yu [1,18], Konstantinos Hatzikotoulas [1], Young-Chan Park[1], Peter Kreitmaier[1,2,3], Georgia Katsoula[1,2,3], Reinhard Berner[4], Kristina Casteels [5,6], Helena Elding Larsson [7,8], Olga Kordonouri[9], Mariusz Ołtarzewski[10], Agnieszka Szypowska[11], Raffael Ott[12], Andreas Weiss[12], Christiane Winkler[12,13], Jose Zapardiel-Gonzalo [12], Agnese Petrera[14], Stefanie M. Hauck [14], Ezio Bonifacio [15,16], Anette-Gabriele Ziegler[12,13,17,19] & Eleftheria Zeggini [1,3,19] ✉

Type 1 diabetes is a chronic, autoimmune disease characterized by the destruction of insulin-producing β-cells in the pancreas. Early detection can facilitate timely intervention, potentially delaying or preventing disease onset. Circulating proteins reflect dysregulated biological processes and offer insights into early disease mechanisms. Here, we construct a genome-wide pQTL map of 1985 proteins in 695 newborn babies (median age 2 days) at increased genetic risk of developing Type 1 diabetes. We identify 535 pQTLs (352 *cis*-pQTLs, 183 *trans*-pQTLs), 62 of which characteristic of newborns. We show colocalization of pQTLs for CTRB1, APOBR, IL7R, CPA1, and PNLIPRP1 with Type 1 diabetes GWAS signals, and Mendelian randomization causally implicates each of these five proteins in the aetiology of Type 1 diabetes. Our study illustrates the utility of newborn molecular profiles for discovering potential drug targets for childhood diseases of significant concern.

T1D is a chronic autoimmune disease often diagnosed during childhood and adolescence. The incidence of the disease is increasing and having T1D is associated with a reduction in quality of life, a shorter life span, and cost-intensive treatment[1,2]. Therapies and strategies to prevent T1D are, therefore, needed.

Advances in genomics and proteomics present new opportunities to better understand disease mechanisms and develop preventive therapies. Both Olink and SomaScan technologies have been employed to measure plasma protein levels in tens of thousands of individuals. By integrating genotype data available for the same

[1]Institute of Translational Genomics, Helmholtz Zentrum München – German Research Center for Environmental Health, Neuherberg, Germany. [2]Technical University of Munich, TUM School of Medicine and Health, Graduate School of Experimental Medicine, Munich, Germany. [3]Technical University of Munich and Klinikum Rechts der Isar, TUM School of Medicine and Health, 81675 Munich, Germany. [4]Department of Pediatrics, University Hospital Carl Gustav Carus, Technische Universität Dresden, Dresden, Germany. [5]Department of Pediatrics, University Hospitals Leuven, Leuven, Belgium. [6]Department of Development and Regeneration, KU Leuven, Leuven, Belgium. [7]Unit for Pediatric Endocrinology, Department of Clinical Sciences Malmö, Lund University, Lund, Sweden. [8]Department of Paediatrics, Skane University Hospital, Malmö/Lund, Lund, Sweden. [9]Kinder- und Jugendkrankenhaus AUF DER BULT, Hannover, Germany. [10]Department of Screening and Metabolic Diagnostics, Institute of Mother and Child, Warsaw, Poland. [11]Department of Paediatric Diabetology and Paediatrics, Medical University of Warsaw, Warsaw, Poland. [12]Institute of Diabetes Research, Helmholtz Munich, German Research Center for Environmental Health, Munich, Germany. [13]Forschergruppe Diabetes e.V. at Helmholtz Munich, Munich, Germany. [14]Metabolomics and Proteomics Core, Helmholtz Zentrum München - German Research Center for Environmental Health, Munich, Germany. [15]Center for Regenerative Therapies Dresden, Technische Universität Dresden, Dresden, Germany. [16]Paul Langerhans Institute Dresden of the Helmholtz Munich at University Hospital Carl Gustav Carus and Faculty of Medicine, Technische Universität Dresden, Dresden, Germany. [17]Forschergruppe Diabetes, School of Medicine, Klinikum rechts der Isar, Technical University Munich, Munich, Germany. [18]These authors contributed equally: Mauro Tutino, Nancy Yiu-Lin Yu. [19]These authors jointly supervised this work: Eleftheria Zeggini, Anette-Gabriele Ziegler. ✉e-mail: eleftheria.zeggini@helmholtz-munich.de

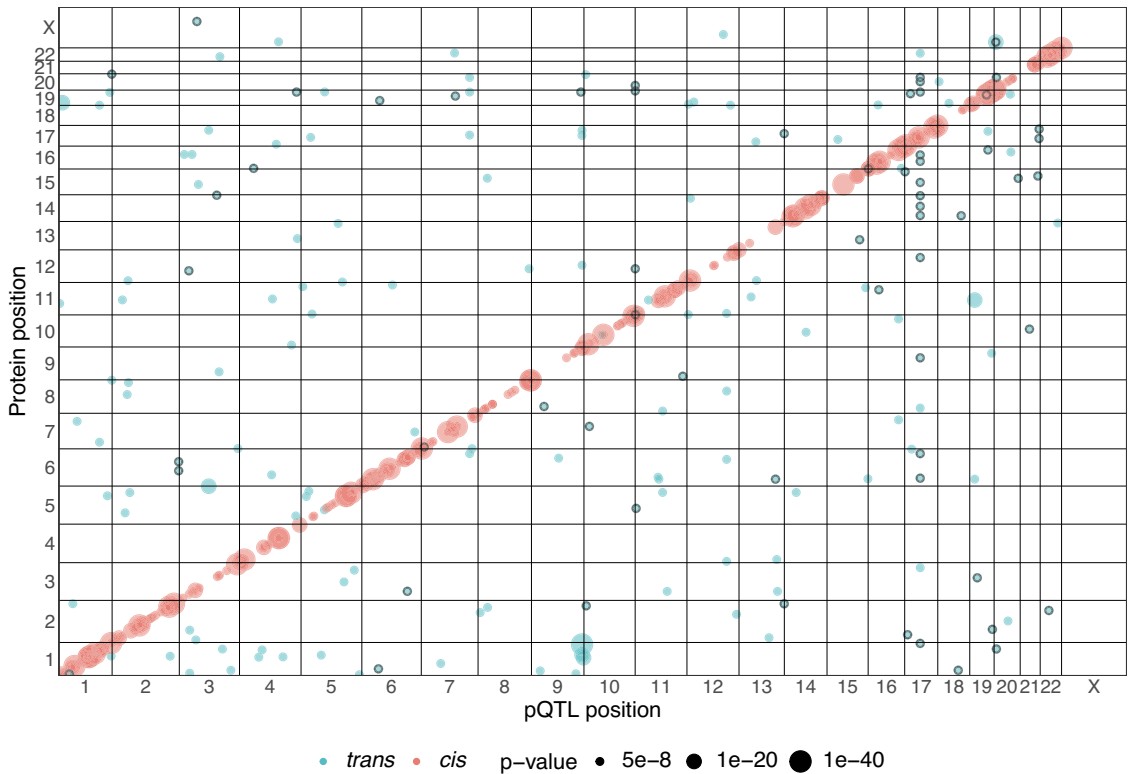

**Fig. 1 | Genome-wide pQTL signals.** Scatterplot of significant pQTL ($p < 5 \times 10^{-8}$) variant location against the position of the protein coding gene's transcription start site. Each dot represents an independent variant. The black point outline identifies pQTLs not previously reported by any of the 46 studies used to define novelty.

individuals, hundreds of genetic variants have been linked to both protein levels (protein quantitative trait loci - pQTL) and complex diseases[3–5]. These findings are important since they provide a genetic anchor to explore potential causal relationships between protein levels and disease risk. A major limitation of proteogenomic studies to date is their focus on adult cohorts. Consequently, the applicability of adult pQTL findings to earlier life stages, such as infancy, remains unproven. Moreover, these population-based studies might have missed disease specific signals.

Coupling genetics with circulating blood proteomics at birth has the potential of identifying stable and infancy-specific pQTLs. In return this might identify early life biomarkers and uncover mechanistic insights of T1D. The Global Platform for the Prevention of Autoimmune Diabetes (GPPAD) has been established to screen for neonates at increased risk of developing T1D[6], and enrol them in clinical trials for early intervention strategies. Here, we have established a protocol for quantifying the levels of circulating proteins from dried blood spots in 695 newborn babies (N. females = 346; 49.78%) enrolled in the GPPAD POInT trial[7] to identify post-natal proteomic signatures at birth and investigate their association with T1D. Imputed genotype data were also available for the same newborns. By generating pQTL profiles and combining these with genetic colocalization and causal inference analyses, we aimed to identify key pathways and proteins involved in T1D. The insights gained from this research could serve as a solid foundation for developing therapeutic targets or repurposing existing drugs to prevent or treat T1D.

## Results

### Newborn blood pQTL map
Protein levels were measured from dried blood spots with the Olink Explore panel. After quality control, 1985 proteins were retained for further analyses. First, we performed genome-wide association analysis (Supplementary Data 1) and identified 535 pQTLs targeting 467

unique proteins (471 Olink assays) at the genome-wide significance threshold of $5 \times 10^{-8}$ (Fig. 1 and Supplementary Fig. 1). Of these, 352 pQTLs are in *cis*, i.e. located within 1MB of the transcription start site of the targeted protein-expressing gene, and 183 are *trans*-pQTL signals (Fig. 1). The *cis* signals were in close proximity to the transcription start site of the protein encoding genes, with an average distance of 40 kb. Olink includes a set of 5 proteins measured on multiple assays, which can be used for quality control. For 2 out of 5 of these proteins (LMOD1 and IDO1), we identified *cis*-pQTL signals which were statistically significant for all three independent assay measurements that passed quality control, suggesting good reproducibility of the results.

We identified 2 *cis*-pQTL signals where the same SNP was also a *trans*-pQTL for a different protein. The *cis* lead variants for CTRB1 (rs72802342) and ASRGL1 (rs72923263) were found to be associated in *trans* with the protein levels of CPA1 and PNLIPRP1, and RNF5, respectively. The IL12B *cis* lead variant (rs6556411) was associated with 2 different Olink IDs, one specific to IL12B and one targeting both IL12A and IL12B, suggesting that both signals are likely due to a IL12B *cis*-pQTL. For 23 proteins, more than one independent *cis* signal was identified, suggesting complex regulatory mechanisms of these protein levels. We replicated pleiotropic loci such as ABO[5], which we found to be associated with the levels of 5 different proteins in *trans*. We also identified a highly pleiotropic locus on the chromosome 17q, associated with the levels of 17 different proteins across 12 chromosomes, involving immune related genes such as the TNF Alpha Induced Protein 2 (TNFAIP2) and the TNF Superfamily Member 10 (TNFSF10).

We compared our pQTL findings with a total of 46 pQTL studies carried out in adults (Supplementary Data 2), including the latest UK Biobank (UKBB) genome-wide plasma pQTL study based on 54,306 individuals[4], and a *cis*-focused proteogenomic analysis of 1180 individuals[3], both of which used the same Olink

panel as the current study. We find that, for the most part, the newborn pQTLs are recapitulated in adult pQTL studies, while 12% appear to be specific to newborn blood samples and/or to children with an elevated risk of developing T1D. Of the 535 SNP-protein pairs, 473 (88%) were previously detected in adults. Of the 62 pQTLs that have not been previously reported, 4 are *cis*-pQTLs, while 58 are in *trans*. The 4 newly-reported *cis*-pQTLs include the Islet Cell Autoantigen 1 (ICA1; pQTL lead SNP: rs7785777), a potential minor T1D autoantigen, AKT Serine/Threonine Kinase 2 (AKT2; pQTL lead SNP: rs4530264), which is involved in insulin signalling[8], HRas Proto-Oncogene, GTPase (HRAS; pQTL lead SNP: rs2061586) and Mitochondrial Ribosomal Protein L28 (MRPL28; pQTL lead SNP: rs3859153). We then investigated if the genes encoding the proteins targeted by the 62 novel pQTL signals were overrepresented in specific pathways. We found a significant enrichment for the KEGG Insulin signalling pathway that was specific for the novel pQTLs (*q*-value = 0.007, Supplementary Data 3 and Supplementary Fig. 2), and which included two of the novel *cis*-pQTL signals (HRAS and AKT2). The replicated signals were enriched for Sphingolipid metabolism, Metabolic pathways and the Viral protein interaction with cytokine and cytokine receptor pathway (Supplementary Data 3 and Supplementary Fig. 2).

## Comparison of newborn pQTLs with adult pQTL and eQTL datasets

The Olink platform uses an affinity assay, which can be affected by protein-coding variants. This can result in genetic associations identified by a change in antibody affinity rather than protein levels. Since gene expression would be less affected by missense variants, and it would not be expected to share the same directionality if the pQTL signal was due to antibody affinity, we queried the GTEx and eQTLGen eQTL databases. We found that all four novel *cis*-pQTLs have been previously associated to the gene expression levels of the genes encoding the pQTL-targeted proteins. Three pQTLs have been identified as eQTLs in GTEx (MRPL28, AKT2 and HRAS) and all four have been reported as eQTLs by eQTLGen, all with the same direction of effect. Next, we queried a recent study which performed pQTL analysis for protein levels measured with both Olink and SomaScan, and which also calculated the correlation of protein measurements between the two technologies from matching samples[4]. In general, replicated and novel pQTL signals showed similar correlation coefficients between the protein levels measured by the two technologies, with a median spearman coefficient of 0.45 and 0.40, respectively. For the novel *cis*-pQTL signals, 2 out of 4 proteins are measured by both technologies. AKT2, which is targeted by 3 different SomaScan probes, showed good correlation between Olink measurement and SomaScan measurement from all 3 probes, ranging from 0.45 to 0.76. However, HRAS showed a very low correlation of 0.03.

We then evaluated if the novel associations showed any difference in minor allele frequency (MAF) between populations and did not find any substantial difference, with a MAF correlation between the studies of 0.99. The novel associations followed the expected relationship between MAF and effect size with no observable difference compared to previously identified pQTLs (Supplementary Fig. 3). We also found high correlation between our study and UKBB effect sizes, with a Pearson correlation of betas of 0.77 (Supplementary Fig. 4). We then queried the full UKBB-pQTL to determine if the novel associations in our study could be identified in the UKBB but did not reach their pQTL genome-wide threshold. We found that, out of 62 novel pQTLs, 16 showed some evidence of association in the UKBB with the same direction of effect. This included 1 of the 4 *cis* signals, targeting MRPL28, which had a *p*-value of $6.5 \times 10^{-7}$ in the UKBB.

Finally, since the proteins targeted by novel pQTLs were over-represented in the insulin signalling pathway, we tested if the protein-SNP pairs identified by our study can be identified as pQTLs in a subset of UKBB participants with self-reported T1D. Out of 53,060 Olink samples, 61 had matching genotype data and self-reported T1D. We replicated 3 *trans*-pQTL results in the T1D subset with the same direction of effect (*p*-value < 0.05; Supplementary Data 4). None of these were nominally significant in the full UKBB dataset (*p*-value > 0.05).

The replicated signals targeted proteins which can be biologically linked to type 1 diabetes, such as the Serine Palmitoyltransferase Long Chain Base Subunit 1 (SPTLC1), a protein involved in the sphingolipid metabolism. For SPTLC1, a previous study found a SNP in the *cis*-SPTLC1 locus associated with T-helper cell proportions in a T1D cohort[9]. A second protein, the IQ Motif Containing GTPase Activating Protein 2 (IQGAP2), has no clear link to T1D from the literature but the targeting SNP (rs112344603) sits 300 kb from the insulin gene promoter. For the third *trans* signal, targeting the WAP, Kazal, immunoglobulin, Kunitz, and NTR domain–containing protein 2 (WFIKKN2), a loss-of-function variant affecting WFIKKN2 circulating protein levels has been associated with HOMA-IR levels[10].

## Colocalization and causal inference with T1D signals

Next, we performed statistical colocalization analysis between pQTLs (both *cis* and *trans* signals) and established T1D GWAS signals from the largest meta-analysis to date[11] (Supplementary Data 5 and 6). Colocalization analysis was performed using the R package Coloc, an established Bayesian method that uses the full summary statistics from the pQTL and GWAS studies, centred around the GWAS lead variant (+/−1 Mb), to calculate the poster probability of the two signals sharing the same causal variant in the locus. We found evidence of a shared causal T1D risk variant with pQTLs for 5 proteins: CTRB1 (PP4 99.7%), CPA1 (PP4 99.7%; *trans*-pQTL), PNLIPRP1 (PP4 99.6%; *trans*-pQTL), APOBR (PP4 92.1%), and IL7R (PP4 68.8%) (Fig. 2). The *cis* signal for CTRB1 and the *trans* signals for PNLIPRP1 and CPA1 shared the same lead variant, rs72802342, and the same colocalization signal with T1D.

The protein levels of CTRB1 and IL7R have been previously shown to highly correlate between SomaScan and Olink measurements, with Spearman coefficients of 0.78 and 0.73, respectively. In addition, for these proteins, the same Olink pQTL signal was identified using SomaScan in adult blood. Altogether, these observations strengthen the validity of our results and reduce the likelihood that the observed signals are solely due to epitope-modifying SNPs. APOBR was not measured by SomaScan.

We then tested for causality of the genetically-regulated, colocalizing protein levels on T1D through Mendelian randomization (MR) analysis. For two-sample Mendelian randomization, the instrumental variables (SNPs) are required to be associated with the exposure (i.e. protein levels). The top independent pQTL for both *cis*- and *trans*-pQTL SNPs identified by GCTA-COJO were therefore used as instrumental variables (IVs) - for each protein, only the lead SNP was used as instrumental variable (Wald ratio). The strength of the association between the IVs and the exposure was further assessed with the F-statistic. All the IVs had an F-statistic >15 and were retained in the analysis. For the outcome, the same T1D GWAS summary statistics utilised for the Coloc analysis were also used for MR. The two-sample MR analysis confirmed that the genetically-predicted protein levels of all Coloc-identified proteins are potentially causally associated with T1D (adjusted *p*-value < 0.05). The colocalization and MR results suggest a potential link between circulating protein levels at birth and the development of T1D later in life (Table 1). However, it is important to note that the MR analysis relied on a single instrumental variable, which limits the robustness of the causal inference due to the inability to fully test for pleiotropy or validate the assumptions underlying the analysis.

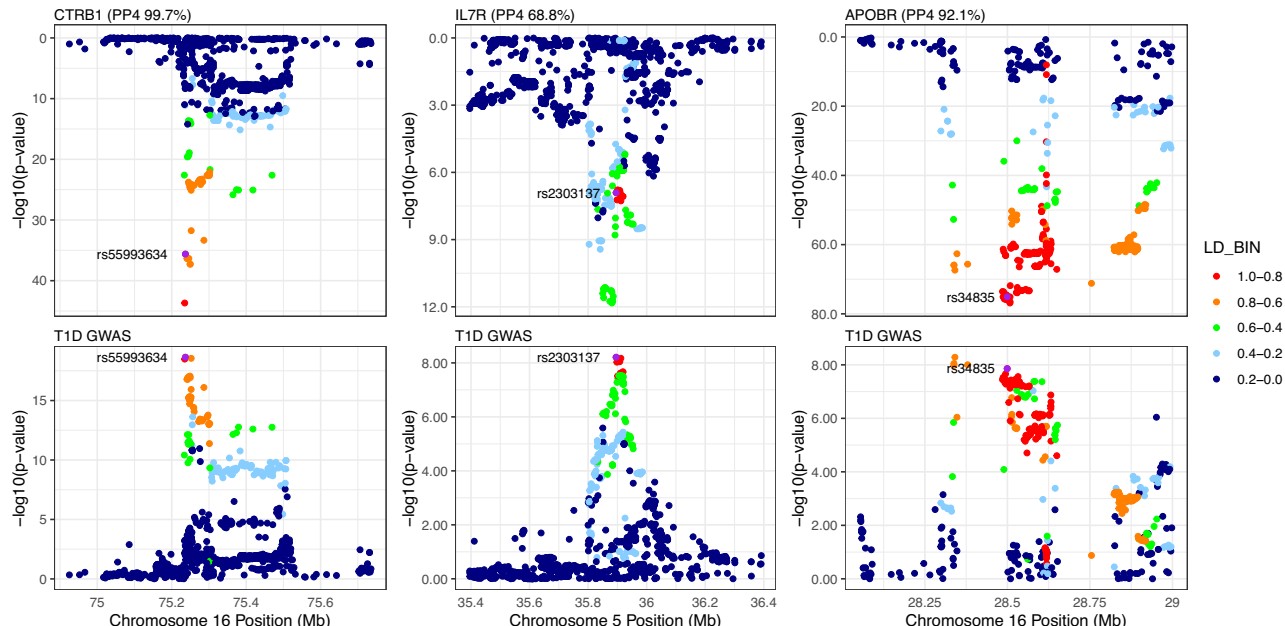

**Fig. 2 | *cis*-pQTL and T1D colocalizing signals.** LocusZoom plots of colocalized proteins for CTRB1, APOBR and IL7R. The top panels show pQTL *p*-values, while the bottom panels show T1D GWAS *p*-values for the same regions for the colocalizing regions. Two independent T1D signals reside in the *cis*-region near the *APOBR* transcription start site, with lead variants rs34835 and rs231972. Each T1D signal (in the +/− 1 Mb window) was conditioned against the other using GCTA-COJO. The summary statistics from the conditional analysis were then used for colocalization. Only rs34835, conditioned on rs231972, showed evidence of colocalization and the corresponding PP4 is reported in the figure.

**Table 1 | Mendelian randomization results for T1D colocalized signals**

| Protein | SNP | Beta | SE | *p*-value | Adjusted *p*-value |
|---------|-----|------|-----|-----------|---------------------|
| **CTRB1** | 16:75234872_rs72802342_A_C | −0.17 | 0.02 | 3.3E-19 | 1.7E-18 |
| **APOBR** | 16:28507644_rs180743_G_C | 0.11 | 0.01 | 1.7E-14 | 8.6E-14 |
| **IL7R** | 5:35874575_rs6897932_T_C | 0.18 | 0.04 | 3.4E-07 | 1.7E-06 |
| **CPA1** | 16:75234872_rs72802342_A_C | −0.32 | 0.04 | 3.3E-19 | 1.7E-18 |
| **PNLIPRP1** | 16:75234872_rs72802342_A_C | −0.41 | 0.05 | 3.3E-19 | 1.7E-18 |

Results of the two-sample MR analysis using the lead pQTL as instrumental variable, protein levels as exposure and T1D (GWAS Catalog accession number GCST90014023) as outcome. The effect size (beta) represents the T1D risk per one standard deviation of the genetically predicted protein levels. Bonferroni adjustment was used to correct the *p*-values. *SNP* single nucleotide polymorphism, *SE* standard error

## Discussion

Here, we have generated a blood pQTL map in newborns at increased risk of developing T1D. We found very good concordance with adult pQTL studies, and identified 62 novel pQTLs enriched for the insulin signalling pathway. A small proportion of these (25.8%) were nominally significant in the full UKBB Olink pQTL data, while 3 were significant only in the UKBB subset with self-reported T1D. We identified a novel *cis*-pQTL signal for the ICA1 protein that could not be replicated in UKBB participants with T1D. The protein is also currently not measured by SomaLogic assays. Expression of *ICA1* has been previously associated with T1D in regulatory T-cells[12] and the expressed protein has also been detected in pancreatic islet cells[13], suggesting a complex multi-tissue effect. Whether the novel associations, that could not be identified by biobank-scale studies in adults, are due to increased risk of T1D in absence of the condition, or to newborns, or a combination of the two, will require future replication.

We identified three T1D-colocalizing pQTLs potentially causally associated with disease aetiology. The colocalizing proteins are related to pancreatic function and insulin signalling. Our findings demonstrate that genetic variants are linked to both protein levels and disease well before the onset of symptoms, such as at birth. This suggests that the direction of effect is likely SNP → protein → T1D, since a reverse temporal association would not be possible with the only exception being if an early pathology is already present at birth. Altogether, these

results raise the possibility of early protein biomarker identification, which can potentially be used to refine the genetic risk score used to enrol babies in the current study. Due to the young age of the children enrolled in the POInT clinical trial, we currently have limited data on the number of children who have developed T1D. Future work will involve analyzing autoimmunity outcome data after a longer follow-up period, to help identify biomarkers relevant to T1D development and intervention efficacy.

The *cis*-pQTL associated with Chymotrypsinogen B1 (CTRB1) resides 18,026 bp upstream of the canonical *CTRB1* transcription start site (TSS), and 6187 bp upstream of the canonical Chymotrypsinogen B2 (*CTRB2*) TSS. In our study, rs72802342 was a *cis*-pQTL for CTRB1, and a *trans*-pQTL for Pancreatic Carboxypeptidase A (CPA1) and for Pancreatic Lipase Related Protein 1 (PNLIPRP1). Evidence for rs72802342 as a *cis*-pQTL for CTRB1, and as a *trans*-pQTL for CPA1[4] and PNLIPRP1[14], has been reported in previous studies in adults (Supplementary Data 1), with the effect allele A being associated with lower levels of proteins in blood. The rs72802342 A allele, which is associated with increased risk of T1D, has also been associated with decreased risk of type 2 diabetes (T2D)[15]. CTRB1, CPA1 and PNLIPRP1 are all pancreatic enzymes, in agreement with previous associations of rs72802342 with pancreatic traits such as pancreatic volume[16] and pancreatic ductal adenocarcinoma[17], and with its colocalization with an accessible chromatin region in the pancreatic acinar cells and islets[11,18]. The locus

has also been previously associated with alternative splicing of *CTRB2*[19]. A previous MR study analysing adult serum protein pQTLs also identified CTRB1 as a biomarker associated with T1D[20]. Here, we provide evidence for this causal association at birth.

The genetic variant (rs180743) associated with the levels of Apo-lipoprotein B48 receptor (APOBR), a macrophage receptor that binds to dietary triglyceride-rich lipoproteins, has been previously associated with Insulin-like growth factor-1 (IGF-1)[21]. IGF-1 levels have been found to be reduced in T1D children and shown to increase following insulin therapy[22,23].

Previous studies have shown that IL-7 and IL7R play a crucial role in the generation of self-reactive T cells in autoimmune diseases[24]. The *IL7R* variant rs6897932 has been shown to affect splicing[25]. The spliced-transcript, lacking exon 6, is preferentially released in soluble form, sIL7R, with a concomitant reduction of transmembrane IL7 receptor. The increase of the exon-skipping isoform, and the resulting increase in sIL7R levels, have been associated with increased risk of T1D[26,27]. *IL7R* variants have also been associated with blood cell counts, including eosinophil, lymphocyte, basophil, and neutrophils[28,29]. PheWAS associations of the *IL7R* pQTL rs6897932 include further immune-related conditions, such as asthma and allergic diseases[30], pollinosis and atopic dermatitis[31], and multiple sclerosis[25], suggesting a pleiotropic effect of the identified pQTL signal. A previous study using SomaScan[32] also identified an IL7R pQTL signal (rs6451229), in weak LD with our lead variant rs6897932 ($R^2 = 0.28$), as strongly colocalizing with T1D GWAS signal from a GWAS with a smaller sample size ($n = 18,856$). For the same individuals ($n = 485$), the Olink-based IL7R pQTL signal, in perfect LD with rs6897932 ($R^2 = 1$) showed a much weaker colocalization probability (PP4 = 16%) compared to the one reported here (PP4 = 68.8%). This is likely the result of the increased sample size of both the GWAS (from 18,856 to 520,580 study participants) and pQTL (43% increase) studies used in the current study, which allowed us to identity a signal replicated using a separate technology, although at a less stringent threshold that the conventional 80%.

The study has several limitations. First, the identification of a causal association of protein levels with T1D is based on statistical analysis and it is constrained by the limited number of instrumental variables. Further work is required to experimentally confirm the involvement of these proteins in the development of T1D. Additionally, the directionality of the MR results should be interpreted with caution, since blood is likely not the effector tissue for the identified pancreatic enzyme proteins and the direction of the pQTL may vary across different tissues. Given that all newborns in the study had an increased risk of developing T1D, associations specific to healthy-newborns may have been overlooked. Post-translational modifications and epitope-modifying variants may affect the binding of the Olink antibodies, making it difficult to distinguish these effects from true pQTL signals. However, despite this technical limitation, the relatively high correlation with protein levels measured with SomaScan, and the fact that the pQTLs have also been previously identified as eQTLs, suggests that the majority of the findings are unlikely to arise from such technical artefacts. For the previously unreported signals, while we provide orthogonal evidence suggesting that many exhibit an association with the same direction of effect in publicly available eQTL and pQTL datasets, this does not constitute direct replication. The limited availability of dried blood spot samples from newborns, combined with the challenges of optimizing the Olink assays for this material, made it infeasible to further replicate the findings in a newborn cohort. Additionally, the small sample size of the UKBB T1D cohort prevented us from determining whether the novel pQTLs are associated specifically with newborns, an increased risk of T1D in absence of the condition, or both. Future studies validating the identified pQTLs in newborns with and without an elevated risk of T1D will be essential to fully understand the genetic regulation of these proteins.

In conclusion, we demonstrate the feasibility and utility of applying high-throughput proteomics on dried blood spots of newborns to identify potentially causal links between circulating proteins at birth and biological regulatory mechanisms relevant to autoimmune outcomes. Multi-omics molecular profiles from dried blood spots could be used to screen children for debilitating diseases and allow for early intervention as well as prevention of disease.

## Methods
### Ethics
The study (GPPAD dried blood spot omics to predict health and autoimmunity outcomes in children) was approved by the ethical committee of the Technical University Munich (Nr. 517/21 S). Consent to donate samples to the GPPAD-biobank was obtained for 770 children, for whom dried blood spots (DBS) samples for DNA extraction, protein measurements, as well as clinical follow-up data were available. The informed consent was obtained in accordance with country-specific guidelines and ethical review board requirements, providing written and verbal information explaining the objectives of the GPPAD study. The work has been carried out in accordance to the criteria set by the Declaration of Helsinki.

### GPPAD Cohort and study samples
Detailed descriptions of the GPPAD cohort recruitment and selection criteria have been described in previous publications[6,33]. Briefly, GPPAD consists of a clinical network of seven clinical trial centres from five countries: Germany, UK, Poland, Belgium, and Sweden. Children with elevated risks for T1D, detected by HLA typing, SNP-based genetic risk score, and first-degree family history of T1D, were screened and recruited from these centres. 1050 children were enrolled in the POInT trial (a randomized double-blind placebo-controlled prevention trial for T1D)[7].

### Genotype analysis and QC
Genotypes of individuals were measured with Illumina Infinium Global Screening Array-24 (GSA) chips. The calling was done using GenCall by Illumina, and the genotypes were mapped to GRCh37/hg19 using online tools from [http://www.well.ox.ac.uk/~wrayner/strand/index.html]. Sample QC and variant QC were carried out. In the first pre-filtering step, samples and variants with a call rate <90% were excluded. Plink 1.9 was used to perform the QC steps[34]. In the sample QC stage, samples were filtered based on call rates (excluded if <95%), heterozygosity rates, performed using two MAFs, MAF > 1% and MAF < 1% (samples excluded if heterozygosity in either category > 3 standard deviations), dubious sex status (samples excluded if there were discrepancies between annotated sex vs. genotypic sex, or if chromosome X heterozygosity between 0.2 and 0.8), identity by descent (IBD) (samples excluded if PI_HAT > 0.9). To determine ethnicity outliers, the sample genotypes were overlapped with genotypes from the 1000 Genomes Project ([http://internationalgenome.org])[35]. Only SNPs that were found in both datasets were used. Multidimensional scaling was performed using plink. Ethnicity outliers were excluded based on visual examination of the first 2 dimensions. For variant QC, exclusion criteria include SNP call rates <98%, and Hardy Weinberg *p*-value (pHWE) $< 1 \times 10^{-4}$. Prior to imputation, variants with strand, position and allele frequency differences were compared to the HRC panel using the script from [http://www.well.ox.ac.uk/~wrayner/tools/]; v4.2.7. Imputation was performed using Sanger Imputation Service ([https://www.sanger.ac.uk/tool/sanger-imputation-service/]).

Eagle2 v2.4 was used to phase the genotypes[36]. PBWT was used for imputation, with the Haplotype Reference Consortium as reference

panel[37,38]. Post-imputation checks were performed using ic v1.10.0 ([http://www.well.ox.ac.uk/~wrayner/tools/Post-Imputation.html]). Variants with MAF < 0.05, pHWE <1 × 10$^{-4}$, and INFO score <30% were removed. The final imputed dataset contained 5,370,094 variants and 695 samples.

## Sample extraction from dried-blood-spots (DBS)

Blood samples were withdrawn from infant heel-prick or umbilical cordon and collected on Ahlstrom Munksjö TFN filter papers. From each blood spot (DBS), two 3 mm ø punches are generated by using the PerkinElmer DBS Puncher equipped with the Software Wallac DBS Puncher and a Punch Head 3.2 mm. The punches were collected in a separate wells of a low-binding 96-well plate (96-well PCR plate full skirt), and subsequently transferred into a low-binding Eppendorf tube for extraction. Each punch was extracted with 25 µl of extraction buffer (1X PBS + 0.05% TWEEN20 + protease inhibitor cocktail tablets from Roche) for 1 h at room temperature. Eluates were stored at −80 °C until usage. Samples were fully randomized in the plates according to gender, study center and storage temperature (−80 °C, −20 °C, RT). The samples were measured using the Olink Explore 3072 panel (Cardiometabolic I and II, Inflammation I and II, Oncology I and II, and Neurology I and II).

## Olink QC

The Olink technology utilises a Proximity Extension Assay in which a pair of protein-targeting antibodies are tagged with unique complementary oligonucleotide probes. Once bound to the target protein, the probes can hybridize to allow DNA amplification. The amplified signal is finally read using next-generation sequencing.

The samples of 770 children from the GPPAD POINT trial were processed by the Helmholtz Munich proteomics core facility. Reports were generated with data for 2941 proteins from the Olink Explore assay which comprises eight panels targeting inflammation (Inflammation I and II), oncology (Oncology I and II), cardiometabolic (Cardiometabolic I and II) and neurological (Neurology I and II) proteins. Average intra-assay variation was 8%, and average inter-assay variation (between-run) was 20%. The NPX values, which is a log2 arbitrary scale unit, were intensity normalized by Olink[39]. For the QC, proteins labelled as a hook protein (a protein whose normalized value has a non-linear relationship with protein quantity in the assay; $n = 286$) or a bimodal protein (a protein with bimodal rather than normal distribution; $n = 3$) were removed. Olink internal quality control was used to further remove poor quality samples and proteins by using the Sample_QC and Assay_QC status. Samples with <500 counts or that deviate from the median value of the Incubation- and Amplification Controls (spiked into each sample) by > +/− 0.3 NPX receive a sample warning status. The median value of the negative control triplicates is also required to be within 5 standard deviations of a predefined value set for each assay or it would receive an assay warning status. Samples with >10% of warning statuses across assays (by either Sample_QC or Assay_QC) were removed ($n = 44$). Proteins with >5% of samples with the warning status were also removed ($n = 668$). Samples with annotated sex that did not match its genotype sex were removed. Samples without corresponding genotypes were removed. The final protein dataset used for data analysis contained 1985 proteins and 695 samples (N. females = 346).

## Power curves

The R package PowerEQTL v0.3.4 was used to generate power curves as a function of minor allele frequency (MAF; Supplementary Fig. 5). For the calculation, the sample size was set to 695 and the significance threshold to 5 × 10$^{-8}$. Since the protein levels were inverse normal transformed, the outcome standard deviation was set to 1. PowerEQTL was also used to estimate the minimum detectable beta for which we had a power of 50% and 80% as a function of MAF. The so calculated curves were then overlayed to the true data which closely followed the power lines. A data point that, for a given MAF, show an absolute beta estimate much smaller than the estimated minimum detectable beta from the power curves would be an indication of likely false positive.

## pQTL

The single-point-analysis-pipeline version 0.0.2 (dev branch) [https://github.com/hmgu-itg/single-point-analysis-pipeline/tree/dev] was used to perform protein quantitative trait locus association (pQTL) analysis between genotypes and circulating protein levels. Covariates including sex, birth year, plate, mean protein expression per sample, and the top 10 genotype PCs were regressed out with R's lm function. The residuals were then z-score transformed and used as traits to test for associations with genotypes.

GCTA version 1.93.2 beta was used to perform mixed linear model association (MLMA)[40]. The GRM function in GCTA was used to estimate genetic relationships between individuals. Plink 1.9 was used to clump the output of GCTA results. GCTA-COJO, an approximate conditional and joint stepwise model selection analysis was then used to calculate independent SNV at each associated locus for each protein. *cis*-pQTLs were defined as variants that lie within 1 Mb upstream or downstream of transcription start site according to the canonical gene's transcription start site using Biomart v.2.50.3 Ensembl hg19 annotation[41]. *trans*-pQTLs were defined as all variants lying outside of the *cis*-pQTL regions. A genome-wide threshold of 5 × 10$^{-8}$ was used to define the threshold for the pQTLs[42].

## Assessment of newborn pQTLs in adult cohorts

To assess if the identified newborn-pQTLs have been previously reported in adults, we built a database of previously reported signals at genome-wide significance (5 × 10$^{-8}$) from 46 genome-wide pQTL studies (Supplementary Data 2) as previously described[42]. Briefly, we collected and pooled the summary statistics of 46 studies that measured the levels of circulating proteins also present in our panel. For each of the study, we collected information such as author, PMID, size of discovery cohort, peak coordinates, UniProt ID, alleles, allele frequencies, effect sizes and direction, mapped gene, *p*-value, and *cis/trans* status. Missing information was manually curated. When required, the genomic coordinates of both the SNP and the protein-encoding gene's transcription start site were lifted over using the liftOver function from the R package rtracklayer. The database for further updated to include a recent study which looked at 2936 unique proteins[3], and the latest UKBB pQTL data based on Olink Explore 3072 panel[4], the same panel used in the current study. First, we determined if the pQTL proteins were previously studied by matching either their protein or gene names. We then looked for the overlap of the lead pQTL SNP in the database with a +/− 1 Mb window around the pQTL signals reported in the current study. A pQTL was declared novel if no genome-wide significant pQTL overlapped the window for a matched protein.

ClusterProfiler[43] v4.12.0 was used to test for enrichment of the pQTL-targeted proteins in the KEGG pathways. The 1985 proteins used in the pQTL analysis were used as background. A Benjamini–Hochberg corrected *p*-value < 0.05 was considered significant.

## Comparison to adult proteogenomic data

Affinity-based assay measurements could be affected by protein-structure-altering variants which would result in a false *cis*-pQTL signal. To determine whether the identified pQTL-targeted proteins are similarly detected across technologies, we retrieved the Olink-SomaScan protein level correlation coefficients from the Eldjarn, Ferkingstad and Lund et al. supplementary materials[4], which calculated the correlation of protein measurements between the two technologies using matching samples. For the 4, novel, *cis*-pQTL signals, we

queried GTEx[44] and eQTLGen[45] to determine if the pQTL SNPs have been previously detected as eQTL for the gene encoding the pQTL-targeted proteins with the same direction of effect following reference allele matching.

We also downloaded the full UKBB pQTL summary statistics, made available by Sun, Chiou, Traylor et al.[5], through Synapse (project ID syn51364943 [https://www.synapse.org/Synapse:syn51364943/wiki/622119]) using the synapser R package. For each OlinkID-SNP pair, the chromosome, position, effect size, effect allele, minor allele frequency and *p*-value information were extracted. Pearson correlation was used to calculate the correlation coefficients between betas and MAFs in the UKBB and newborn datasets. The full UKBB pQTL summary statistics were examined to determine whether previously unreported pQTLs showed evidence of association in the UKBB while not reaching their study significance threshold.

To assess if the novel associations could be identified in adult individuals with T1D, we accessed genotype (version 3) and Olink (release 9) data for 61 UKBB study participants with self-reported type 1 diabetes. For the Olink dataset, only time point 0 was used. For the matching OlinkID-SNP pair, we used linear regression in R to calculate the genotype association with inverse normal transformed NPX values. Sex, age, mean NPX and time between sample collection (field 3166) and data generation were used as covariates.

### Colocalization

To see if the circulating blood protein levels in newborn blood and T1D have shared causal variants, colocalization was performed between protein QTL summary statistics and T1D GWAS results. Summary statistics from the most recent T1D GWAS were downloaded from the NHGRI-EBI GWAS catalogue (accession number GCST90014023)[11]. Regions ± 1MB of 136 independent T1D GWAS signals were used to check for overlap with 535 pQTLs of this study. The Coloc.fast function from [https://github.com/tobyjohnson/gtx/blob/526120435bb3e29c39fc71604eee03a371ec3753/R/coloc.R] was used for the analysis. The colocalization of the signal in the locus was defined as a posterior probability of sharing the same causal variant at (PP4) > 80%. A PP4 > 60% was also considered as probable for colocalization. As 2 reported T1D risk variants were in the ± 1 MB region of the start site of APOBR, we conditioned each T1D signal (in the +/− 1 Mb window) against the other using GCTA-COJO. The summary statistics from the conditional analysis were then used for colocalization.

### Mendelian randomization

The R package TwoSampleMR v.0.5.6[46] was used to perform Mendelian randomization as a complementary step for the proteins whose pQTL signals colocalized with T1D GWAS signals. Since only the lead pQTL for each *cis* or *trans* signal for each of the proteins was used as instrument, the Wald ratio test was performed. For the outcome data, the utilised T1D GWAS study (accession number GCST90014023) is the result of inverse-variance weighted meta-analysis of 9 European cohorts. The meta-analysis has a combined sample size of 18,942 patients with T1D and 501,638 controls. The reported summary statistics are in the log-odds scale and represent the unit increase/decrease risk of having T1D. The lead pQTL variants from the COJO conditional analysis (*p*-value < $5 \times 10^{-8}$) were used as instrumental variables (Table 1). Betas and standard errors were available for all the instrumental variables from the T1D summary statistics (Supplementary Data 7). The F-statistic, defined as $beta^2/se^2$, was used to determine the strength of the association between IVs and the exposure. All the tested IVs had an F-statistic >15 (Supplementary Data 7). The effect size of the Wald ratio test represents the T1D risk per one standard deviation of the genetically predicted protein levels. *P*-values were adjusted for multiple testing correction using the Bonferroni method and a corrected *p*-value < 0.05 was considered significant. The STROBE-MR checklist is available as Supplementary Data 8.

### Reporting summary

Further information on research design is available in the Nature Portfolio Reporting Summary linked to this article.

### Data availability

The type 1 diabetes GWAS summary statistics used for colocalization analysis can be obtained from the GWAS catalog using the accession number GCST90014023. The summary statistics for the significant pQTLs, and the results from colocalization are provided in the supplementary Data. The full pQTL summary statistics are available for download from the Type 1 Diabetes Knowledge Portal [https://t1d.hugeamp.org/] under the following links: Cardiometabolic:https://api.kpndataregistry.org/api/d/36XKce Cardiometabolic II: https://api.kpndataregistry.org/api/d/Anm8En Inflammation: https://api.kpndataregistry.org/api/d/KH5gTM Inflammation: II https://api.kpndataregistry.org/api/d/9U8Q82 Neurology: https://api.kpndataregistry.org/api/d/FVm5B8 Neurology: II https://api.kpndataregistry.org/api/d/GASY1U Oncology: https://api.kpndataregistry.org/api/d/FBJuLa Oncology: II https://api.kpndataregistry.org/api/d/4JZc7C.

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

## Acknowledgements

We thank the study participants and their parents for their continued support and enthusiasm. This research has been conducted using the UK Biobank Resource under application number 10205. We acknowledge the hard work and dedication of the study teams (postdoctoral scientists, physiologists, research fellows, nurses, technicians, and clerical staff). We are grateful to Iris Fischer, Lorraine Southam, Ozvan Bocher and Andrei Barysenka for their support. The current work is funded by Bundesministerium für Bildung und Forschung (grant No. 01KX1818) and The Leona M. and Harry B. Helmsley Charitable Trust (grant no. G-2018PG-T1D022; G-2018PG-T1D023; G-2103-05036).

## Author contributions

M.T. and N.Y. performed the statistical analyses and wrote the manuscript. A.Z. and E.Z. conceived and planned the study, and supervised the work. K.H. performed the quality control of genetic data. Y.P., P.K. and G.K. contributed to the quality control and analysis of Olink data. A.P. and S.H., generated the Olink data from dried blood spots. R.B., K.C., H.E.L., O.K., M.O., R.O., A.S., A.W., C.W., J.Z., and E.B. contributed to the sample collection. All authors provided critical feedback and helped shape the research, analysis, and manuscript.

## Funding

## Competing interests

The authors declare no competing interests.
