## [Transparent Peer Review file · Nature Communications]

Genetics of circulating proteins in newborn babies at high risk of type 1 diabetes

Corresponding Author: Professor Eleftheria Zeggini

Version 0:

Reviewer comments:

Reviewer #1

(Remarks to the Author)

The present study reports a GWAS with almost 2,000 proteins measured in dried blood spots from 700 newborns that were at increased risk of developing T1D. The authors report 535 pQTLs, 62 of which have not been reported before. They show that pQTLs for CTRB1, APOBR, IL7R, CPA1, and PNLIPRP1 colocalize with T1D GWAS signals and implicate these five proteins in the aetiology of T1D using Mendelian Randomization. The paper is concise but to the point in that it reports novel pQTLs that are of clinical relevance.

I have only one major point to comment on, and that is the use of the term “causal” in relation to the MR analysis. Two sample MR using Wald ratios does not show causality on its own. The authors have not shown that all assumptions of MR are valid, i.e. it is known that the Olink platform may be prone to epitope changing variants. It is possible that genetic variants change the protein binding of the antibodies without affecting the protein levels. The computational result of the MR would be the same, but the conclusion that the levels of the affected proteins are causal for changes in the outcome is certainly not true in such cases. Also, proteins are measured in blood, whereas their effects on the outcomes may be through processes located in some organs. There is thus the possibility of horizontal pleiotropy between the same protein expressed in different tissues and their readout in blood that cannot be ruled out. It is even possible that a genetic variant increases protein levels in blood to compensate for lack of protein in some organs. In such cases the directionality of the MR estimate could be the inverse of the true effect and misguide clinical intervention.

Reviewer #2

(Remarks to the Author)

Yiu Lin, Titino et al provide a study that performs a protein GWAS of Olink proteins in newborns at a higher risk for T1D. They identify both cis and trans pQTLs and attempt to put their findings into context with published GWAS of proteins in adults. Additional studies such as colocalization and Mendelian randomization have been performed in effects to explore whether the pQTLs may have an effect on T1D.

This is an interesting concept. However, I found the manuscript difficult to read, given the lack of headings of sections (abstract, intro, results, discussion). I also found the manuscript to be lacking detail, especially in the methods and results, therefore this manuscript needs more work to be publishable. There are some questions I have about some of the approaches used in the methods. I also think the interpretation of results may not be very accurate. Specific details are below.

Introduction

- Very brief introduction with 5 references. Authors need to give a more thorough overview of the current field.

Methods

- Even though a reference to additional study information has been provided, factors such as the age of the children is important for the current study and therefore should be included in the manuscript. How do they define newborn?
- If all are high risk of T1D, and no controls, will important signals for T1D be missed in the GWAS?
- Supp Table 2 lacks detail on each study and why they are being highlighted.
- Line 222 - what does problematic proteins and problematic samples mean? What is this based on? Is it limit of detection?
- Line 262, $h4$ of > 0.6 defined as probable colocalization. This may be a bit too lenient.
- From line 267 - description of Mendelian randomization far too brief. Which GWAS were used for exposure and outcome? What p value for SNPs?
- Did authors perform any power calculations for this GWAS? Small sample size used.

Results/discussion

- Figure 1 - key for p value threshold unclear, all points for p value thresholds look the same. Points overlaying which changes the transparency of points. Maybe use size of the points for p value scale.
- When referencing Figure 1 - could make more specific references to Figure 1A/1B.
- Figure 2 - Axis labelled protein is confusing, how are these sorted?
- Authors could provide more of a summary of the pQTLs identified more broadly - they mentioned that one SNP is cis for one protein and trans for another. What about all the other pQTLs from the GWAS? How many cis pQTLs were also trans pQTLs for other proteins?
- From line 118 - if the SNP for CTRB1 also associates with other proteins, does this suggest that there may be pleiotropy? Is the MR estimate truly estimating an effect of CTRB1 on T1D?
- Unclear where the discussion starts
- Limitations need to be addressed
- Sentence on 149 does not seem justified - you have identified pQTLs within a high risk group, but have not shown that the levels of this protein are different compared to controls.

Other

- No conflict of interest or author contributions

Reviewer #3

(Remarks to the Author)

Genetics of circulating proteins 1 in newborn babies at high risk of type 1 diabetes
Yu et al.

Summary

The authors construct a genome-wide pQTL map of 1985 proteins in 695 newborns at high risk of T1D, showing 535 pQTLs (66% cis) in 437 unique proteins of which 62 are novel. Five pQTL, selected from overlap with T1D associated variants, show colocalisation with T1D susceptibility loci with Mendelian Randomisation implicating them in the aetiology of T1D. The novel variants identified are difficult to interpret given the technical concerns over the platform (affinity impacted by protein-coding variants) and a lack of validation. Most pQTL identified are previously reported and two of the three 'causal' variants identified (IL7R, CTRB1) have supportive evidence on another platform as pQTL. The principle novelty in what is shown is their colocalisation with T1D risk variants and the MR analysis indicating a causal role in T1D. Other pQTL are effectively validation of reported associations in newborns or unvalidated new pQTL in a distinct and much smaller cohort. There is clear discussion of previous functional work on the 2 causal variants but no new data presented. The authors claim that their results may allow inclusion of protein quantification as biomarkers for risk stratification or for drug target identification but it isn't clear what this might add and no new data is presented.

Comments:

Methods

Use an established screening platform (GPPAD) to identify high risk children and an ongoing therapeutic trial of oral insulin nested within that platform.

Use dried blood spot from newborns to quantify 1985 proteins using OLink platform.

pQTL analysis on those proteins with colocalization and causal inference (mendelian randomisation) to investigate link to T1D.

No discussion of why restricted to MR on cis only variants with trans variants excluded.

Not clear how multiple correction was handled during the identification of 'significant' coloc for the 5 variants (and not explicit over how many variants tested from the meta-analysis).

Results

Fig 1A is not a very helpful way to present the findings as much of what is shown is not clearly interpretable. Little actual data is presented - for example we are not shown the proteins tested, which overlap with the adult cohort and which are 'new'.

The colour legend for Fig1B appears to relate to cis/trans category only and not the magnitude of significance as presented.

It would also be relevant to show which pQTL that are reported elsewhere were not detected amongst the 467 proteins with a pQTL in newborns. It is claimed that pQTL are largely stable between newborns and adults, although we don't see what % of the 1985-467 = 1518 proteins that don't show a pQTL here, did show one in adults (I accept the power of detection is very different, but that is a relevant limitation to the discovery here).

We aren't told how the 1985 proteins are selected (presumably based on availability through Olinks panels, but those are more extensive than the proteins tested here).

The link between variants and detection is clearly an issue for all new and potentially for previously-reported pQTL also. Reader cannot easily define what is valid given the data presented. The issue potentially relates to any O-link quantified protein that is not validated with a secondary method, and that appears to include most of what is presented here.

Minimal discussion of the 'New' pQTL and why they are detected here and not elsewhere?

No description of possible mechanism, although not likely to be simple given expression of ICA1 in pancreas and in Treg (with expression here presumably dictated more by the latter).

Validation of most pQTL is not necessary (as they are previously reported and hence this study is validation, only in newborns). However, it would be necessary to validate the 'new' findings, both technically (with another platform) and in another cohort.

Only 2 of the three cis pQTL run validated in Somalogic data and no attempt to validate the association with a secondary technology.

Trans-pQTL not run with mendelian randomisation and not made clear why the restriction to cis-variants was made.

CTRB1 variant has several previously-described associations including pancreatic volume and T1D. The principal novel result here is the coloc and MR analysis suggesting causality. No new data on the putative mechanism is presented.

IL7R - previous functional characterisation of the variant but nothing new presented here.

Central to the discussion is the proposal that new pQTL signals present at birth but not later could indicate an early pathology. This is very speculative, particularly as there is no validation of the 'new' pQTL signals, with no colocalisation or MR analysis of them.

Similarly, it is claimed (in both abstract and discussion) that the proteins could be used as supportive biomarkers for early diagnosis, supporting GRS. However, GRS already have access to the SNP variants associated with T1D on which these pQTL are based and no data is presented to support their utility as biomarkers or therapeutic targets (although they could have been considered as either before this study given the previously demonstrated SNP association and, in some cases, demonstrated protein level association with disease).

Version 1:

Reviewer comments:

Reviewer #1

(Remarks to the Author)

The author responded all concerns

Reviewer #2

(Remarks to the Author)

The manuscript by authors has improved since the first version. I still have a few issues with some of the methodology and there are some outstanding queries after authors provided their previous response. It would be really helpful to have the changes made to the paper copied into the response to reviewers document, along with line numbers of where changes were implemented.

For example, authors responded to some of my queries, but I still have a few questions:

(1) In my previous comment I indicated that using $h4$ of > 0.6 defined as probable colocalization is not appropriate. I'm familiar with the papers the authors have shared and understand (and agree) with their sentiments, however I have two points: (1) categorising probabilities would be akin to categorisations of p-values where, for example, < 0.05 indicates evidence of an effect 0.05-0.1 indicates suggestive evidence, this should be avoided and instead the probabilities (as with p-values) should be interpreted as a continuum in light of the surrounding evidence; (2) the calculation of the probability of colocalization conditional on the presence of a causal variant is dependent upon $p12$. As $p12$ changes so will this probability, therefore signals which are not robust may show larger probabilities simply because of $p12$ and window size. I am happy with the analyses but suggest the authors just rephrase such that the interpretation is given in light of other evidence and as a continuum, which is susceptible to priors and window size. As a side note, the justification behind the choice of probably colocalization of $h4$ between 0.6 and 0.8 was that other papers had done the same. This is not a valid justification of why you selected certain methods - there are plenty of papers which repeat poor methods, which does not

make it correct.

(2) In response to my comment about the power calculation, authors again cited other papers that had used similar sample sizes. A more thorough approach is performing the power calculation. Just because there are smaller GWAS out there doesn't mean they are well performed studies.

(3) The Olink QC section has been updated following my previous comment. A description of the Olink technology would be useful here. Based on the updated reporting of whether proteins or samples were excluded - the defined criteria seem very strict (where samples with >10% warnings across assays were removed or proteins with >5% missingness across samples were removed). The latter removes 668 proteins, where there still may be some important information. I think it would be better to include more proteins here and to allow more missingness (for example, exclude proteins where there is >20% missingness for a protein). Including more proteins, but being fully transparent about missingness of all proteins and warnings may be a better approach. Or alternatively, using multiple imputation. Why were these criteria chosen?

(4) Authors have attempted to add more detail to their Mendelian randomization analysis. However, the extra Mendelian randomization methodological detail is not as I would expect. The individual functions used in the TwoSampleMR package are not necessary information. As mentioned previously, giving a clear overview of the GWAS that performed was performed to give you your summary statistics etc are important. I would advise authors fill out and upload the STROBE-MR checklist to ensure they are following recommended guidelines. I would also advise that authors make their code for analyses available on GitHub or Zenodo so that analyses are reproducible.

Some additional queries I have

- The "Comparison to adult proteogenomic data" section from line 534 is not written very clearly. Please ensure when writing methods that you are indicating why you have performed an analysis. For example, why were correlation coefficients used? I know it is to see whether they are measured similarly across technologies (after reading the response to reviewers), but this is not clear in the main manuscript in the methods.
- Again, categorisation of results into $p < 0.05$ and $p > 0.05$ as significant and non-significant, respectively, should not be used.
- Instead, strength of evidence should be reported.
- Olink having same protein on multiple assays is more of a technical control rather than validating the pQTL, as it is still performed in the same sample?

Reviewer #3

(Remarks to the Author)

Thank you to the authors for their thoughtful response to the queries raised. I appreciate their efforts to provide additional supportive data to bolster their conclusions and the additional interpretation of the results included in discussion.

For me, the biggest issue remains that the 'new' pQTLs presented are not validated, either in another cohort or with another technology (with significant known caveats with SNP variants influencing protein detection for the methods presented).

I fully appreciate the challenge of obtaining another cohort comparable to this, but the analyses presented do not show validation (with concordance of direction of effect not the same thing).

Version 2:

Reviewer comments:

Reviewer #2

(Remarks to the Author)

Reviewer #3

(Remarks to the Author)

The authors have conceded that they do not replicate the new findings presented and this has now been acknowledged in the manuscript.

Reviewer #4

(Remarks to the Author)

I have only reviewed the MR section of the manuscript as per my expertise.

This MR appears to be appropriately conducted however there are some details lacking and the results are not surprising given how the results were selected to be included in the MR.

Regarding the details of the MR the authors should be explicit that they use only a single SNP for each MR (and not multiple

SNPs from the same region). The strength of the association between the SNP and the exposure should be calculated and stated using an F-statistic.

My second comment regards more the use of MR in this setting and how these exposures were selected to be included in the MR. MR is most definitely not an orthogonal test for causality to the colocalisation that was performed in the previous step (as is currently stated in the methods). Both analyses are using the same data to test the correlation between the association of a SNP with the exposure and T1D, but in different ways. As only exposures for which the coloc showed strong evidence of colocalization with T1D it is unsurprising that they all show strong effects in the MR as they have only selected the exposures where we would expect to see an effect. The language around the interpretation of these results in the paper currently is too strong for what the results actually show.

A limitation of all MR studies is that the assumption of no pleiotropic effects of the SNPs on the outcome. In this case it is not possible to test this assumption and so no further analysis is needed. However the limits on the ability of MR to test for causality and the assumptions required should be stated in the final paragraph of the results.

Finally, how should the beta's from the MR be interpreted? What scale are these results on? Are they log odds ratios? If so the odds ratios should also be reported to enable interpretation of the potential magnitude of the effect estimated by the MR.

Version 3:

Reviewer comments:

Reviewer #4

(Remarks to the Author)

The authors have addressed all of my concerns. I have no further comments.

Genetics of circulating proteins in newborn babies at high risk of type 1 diabetes

We thank the reviewers for their comments, which have helped improve the manuscript. We provide a point-by-point response below.

Reviewer 1

The present study reports a GWAS with almost 2,000 proteins measured in dried blood spots from 700 newborns that were at increased risk of developing T1D. The authors report 535 pQTLs, 62 of which have not been reported before. They show that pQTLs for CTRB1, APOBR, IL7R, CPA1, and PNLIPRP1 colocalize with T1D GWAS signals and implicate these five proteins in the aetiology of T1D using Mendelian Randomization. The paper is concise but to the point in that it reports novel pQTLs that are of clinical relevance.

I have only one major point to comment on, and that is the use of the term “causal” in relation to the MR analysis. Two sample MR using Wald ratios does not show causality on its own. The authors have not shown that all assumptions of MR are valid, i.e. it is known that the Olink platform may be prone to epitope changing variants. It is possible that genetic variants change the protein binding of the antibodies without affecting the protein levels. The computational result of the MR would be the same, but the conclusion that the levels of the affected proteins are causal for changes in the outcome is certainly not true in such cases. Also, proteins are measured in blood, whereas their effects on the outcomes may be through processes located in some organs. There is thus the possibility of horizontal pleiotropy between the same protein expressed in different tissues and their readout in blood that cannot be ruled out. It is even possible that a genetic variant increases protein levels in blood to compensate for lack of protein in some organs. In such cases the directionality of the MR estimate could be the inverse of the true effect and misguide clinical intervention.

We appreciate the comments regarding the potential influence of epitope-changing variants on the three cis-pQTL signals and the use of the term "causal". We agree that antibody affinity due to epitope changes could contribute to these signals. In order to address this concern, we have now queried the correlation between protein levels measured by two independent technologies, Olink and SomaScan, as reported in a recent study

Eldjarn, G.H., Ferkingstad, E., Lund, S.H. et al. Large-scale plasma proteomics comparisons through genetics and disease associations. *Nature* 622, 348–358 (2023).
<https://doi.org/10.1038/s41586-023-06563-x>

We found that two out of the three cis-targeted proteins -CTRB1 and IL7R- were measured by both technologies. Both showed high correlation with Spearman coefficients of 0.73 and 0.78 for IL7R and CTRB1, respectively. In addition, the Olink pQTL signals for both CTRB1 and IL7R were also identified using SomaScan measurements. The high concordance between these two distinct measurement technologies strengthens the validity of our results and reduces the likelihood that the observed signals are solely artifacts of antibody binding issues. We have incorporated this new comparison into the revised manuscript. We also acknowledge that blood may not represent the primary tissue of effect for the identified pQTL signal colocalizations, particularly for CTRB1, which is a pancreatic enzyme, and that the directionality needs to be interpreted with caution. We have extended the discussion to include this among the limitations of the study and have rephrased the text throughout. We also clarify in the text that the causality inferred from our MR analysis is statistical. While our findings provide strong evidence for an association, the biological causal role of these proteins in T1D pathogenesis requires further experimental validation.

Reviewer 2

Yiu Lin, Titino et al provide a study that performs a protein GWAS of Olink proteins in newborns at a higher risk for T1D. They identify both cis and trans pQTLs and attempt to put their findings into context with published GWAS of proteins in adults. Additional studies such as colocalization and Mendelian randomization have been performed in effects to explore whether the pQTLs may have an effect on T1D.

This is an interesting concept. However, I found the manuscript difficult to read, given the lack of headings of sections (abstract, intro, results, discussion). I also found the manuscript to be lacking detail, especially in the methods and results, therefore this manuscript needs more work to be publishable. There are some questions I have about some of the approaches used in the methods. I also think the interpretation of results may not be very accurate. Specific details are below.

We have extensively re-written and formatted the manuscript, which is now divided into sections with headings and sub-headings. We have also modified the text to include more information in all sections. We hope the new version of the manuscript is now easier to read.

Introduction

- Very brief introduction with 5 references. Authors need to give a more thorough overview of the current field.

Following the reformatting and extensive inclusion of detail, the introduction has also been extended.

Methods

Even though a reference to additional study information has been provided, factors such as the age of the children is important for the current study and therefore should be included in the manuscript. How do they define newborn?

For GPPAD participants included in this study, the median sampling age was 2 days and this information is now included in the manuscript.

If all are high risk of T1D, and no controls, will important signals for T1D be missed in the GWAS
The newborns enrolled in the GPPAD study have a >10% genetic risk of developing multiple beta-cell autoantibodies by the age of 6 years. The estimated prevalence of T1D in these high-risk infants was estimated to be 1.1%. Given the relatively low estimated prevalence, although much higher than the general population, we expect that T1D GWAS signals affecting the risk of developing the condition would remain detectable, albeit with likely reduced effect sizes.

Therefore, while it is possible that these signals did not reach genome-wide significance, we identified colocalization signals with T1D that were previously missed by Olink studies but identified in SomaScan pQTL studies, such as IL7R. On the other hand, pQTL signals specific for T1D would be more visible in this cohort. In support of this, we have added a new analysis in the revised manuscript that focuses on a subset of UKBB study participants with T1D, which allowed us to replicate three T1D-specific signals that were not significant in the full UKBB cohort.

Supp Table 2 lacks detail on each study and why they are being highlighted.

We have provided more information about the database in the manuscript and with reference to the original publication which contains the full description. We have also included more information about each study in Supplementary Table 2.

Line 222 - what does problematic proteins and problematic samples mean? What is this based on? Is it limit of detection?

We apologize for the vague description. We have now described in more detail the definitions for exclusion in the methods and have also added the number of excluded samples and proteins. Specifically, “problematic” was referring to the manufacturer’s assay and sample QC warnings.

Line 262, H_4 of > 0.6 defined as probable colocalization. This may be a bit too lenient.

The H_4 threshold for colocalization can vary across studies, with recent examples using thresholds as low as 0.5

Gudjonsson, A., Gudmundsdottir, V., Axelsson, G.T. et al. A genome-wide association study of serum proteins reveals shared loci with common diseases. *Nat Commun* 13, 480 (2022). <https://doi.org/10.1038/s41467-021-27850-z>

Oliva, M., Demanelis, K., Lu, Y. et al. DNA methylation QTL mapping across diverse human tissues provides molecular links between genetic variation and complex traits. *Nat Genet* 55, 112–122 (2023). <https://doi.org/10.1038/s41588-022-01248-z>

Further review of IL7R colocalization results shows that $H_4/(H_4 + H_3)$, which represents the probability of colocalization conditional on the presence of a causal variant, as described by

Zuber, V., Grinberg, N.F., Gill, D. et al. Combining evidence from Mendelian randomization and colocalization: Review and comparison of approaches. *The American Journal of Human Genetics*, 109, 5 (2022). <https://doi.org/10.1016/j.ajhg.2022.04.001>

is 0.68 and that $H_4/H_3 > 2$, indicating that the probability of the two traits sharing the same causal variant is twice as high as the probability of the two traits having two independent causal variants. However, we acknowledge that we cannot fully exclude the latter given that H_3 is non-zero ($H_3 = 0.16$). We believe these results are still noteworthy, and we have revised the manuscript to categorize the results as having high support ($H_4 > 0.8$) and medium support ($H_4 > 0.6$).

From line 267 - description of Mendelian randomization far too brief. Which GWAS were used for exposure and outcome? What p value for SNPs?

The methods section of the MR analysis has now been extended with substantially higher level of detail.

Did authors perform any power calculations for this GWAS? Small sample size used.

We agree with the reviewer that, compared to biobank-scale projects, the sample size is modest. However, it is larger or only moderately smaller than other recent pQTL studies such as

Gilly, A., Park, YC., Png, G. et al. Whole-genome sequencing analysis of the cardiometabolic proteome. *Nat Commun* 11, 6336 (2020). <https://doi.org/10.1038/s41467-020-20079-2> – sample size = 1328

Koprulu, M., Carrasco-Zanini, J., Wheeler, E. et al. Proteogenomic links to human metabolic diseases. *Nat Metab* 5, 516–528 (2023). <https://doi.org/10.1038/s42255-023-00753-7> – sample size = 1180

Steinberg, J., Southam, L., Roumeliotis, T.I. et al. A molecular quantitative trait locus map for osteoarthritis. *Nat Commun* 12, 1309 (2021). <https://doi.org/10.1038/s41467-021-21593-7> – sample size = 115

Among these, Koprulu *et al.* already demonstrated that smaller cohorts can still contribute to the identification of new biologically relevant proteogenomic links to disease. We hope that the reviewer will agree with us that this is the first proteogenomic study from newborn dried blood spots (median of 2 days) and, therefore, not comparable to previous studies on easily accessible tissues (blood) from adults.

We have also added a deep comparison of our results to the extended UKBB pQTL data and have also generated new data on a subset of UKBB participants with T1D. We have shown that one of our novel *cis*-pQTL signals is also nominally significant in the full UKBB cohort but did not reach genome-wide significance ($p\text{-value} = 6.5 \times 10^{-7}$) and that 3 signals are specific to the T1D-UKBB subgroup.

Therefore, despite the smaller sample size, our findings provide novel insights on the genetic regulation of protein levels in this valuable newborn cohort. To further address power concerns, we have also generated power curves and overlaid them to our results (supplementary figure 3) demonstrating that, given the observed effect sizes, we have sufficient power to detect the reported novel pQTLs.

Figure 1 - key for p value threshold unclear, all points for p value thresholds look the same. Points overlaying which changes the transparency of points. Maybe use size of the points for p value scale. The new figure included in the manuscript now uses the point size instead of transparency. We believe the suggestion improved the readability of the figure and we thank the reviewer for the suggestion.

When referencing Figure 1 - could make more specific references to Figure 1A/1B. Following the reformatting of the manuscript, Figure 1A has been moved to the supplementary materials and referencing Figure 1B is no longer needed.

Figure 2 - Axis labelled protein is confusing, how are these sorted?
The proteins used in Figure 1A are alphabetically sorted. This is now explained in the figure caption.

Authors could provide more of a summary of the pQTLs identified more broadly. We have included more general information about the identified pQTLs, such as the number of *cis*-pQTLs regulating protein levels in *trans* and distance of lead *cis*-pQTLs from their gene target transcription start site.

They mentioned that one SNP is *cis* for one protein and *trans* for another. What about all the other pQTLs from the GWAS? How many *cis* pQTLs were also *trans* pQTLs for other proteins?
As also suggested in the comment above, we have now added an extended summary of the pQTL results, which also includes the number of *cis* signals which were also *trans* for other proteins.

From line 118 - if the SNP for CTRB1 also associates with other proteins, does this suggest that there may be pleiotropy? Is the MR estimate truly estimating an effect of CTRB1 on T1D?
The regulatory mechanisms of *trans*-pQTL signals are not known. For eQTLs, which we might assume follow the same mechanisms, they are thought to be at least partially mediated through *cis* effects

Yang, F., Wang, J., The GTEx Consortium et al. Identifying *cis*-mediators for *trans*-eQTLs across many human tissues using genomic mediation analysis. *Genome Res.* 27, 1859-1871 (2017).
<https://doi.org/10.1101/gr.216754.116>

If that is the case for this locus, the *trans*-pQTL signals would be mediated through *CTRB1 cis*-pQTL. The observation that the *cis* and *trans* colocalizing signals share that same set of SNPs would indeed support this assumption but future experimental studies will be needed to confirm it.

Unclear where the discussion starts.

The whole manuscript has been re-formatted, and a clear Discussion section is now included.

Limitations need to be addressed.

The Discussion now includes a paragraph detailing the limitations of the study.

Sentence on 149 does not seem justified - you have identified pQTLs within a high-risk group, but have not shown that the levels of this protein are different compared to controls.

We agree with the reviewer and the sentence has now been modified:

“Whether the novel results that could not be identified by biobank-scale studies in adults, are due to increased risk of T1D in absence of the condition, or to newborn age, or a combination of the two, will require future replication.”

No conflict of interest or author contributions

We have now added the conflict of interest and author contributions statements to the manuscript.

Reviewer 3

The authors construct a genome-wide pQTL map of 1985 proteins in 695 newborns at high risk of T1D, showing 535pQTLs (66%*cis*) in 437 unique proteins of which 62 are novel. Five pQTL, selected from overlap with T1D associated variants, show colocalisation with T1D susceptibility loci with Mendelian Randomisation implicating them in the aetiology of T1D.

The novel variants identified are difficult to interpret given the technical concerns over the platform (affinity impacted by protein-coding variants) and a lack of validation. Most pQTL identified are previously reported and two of the three 'causal' variants identified (*IL7R*, *CTRB1*) have supportive evidence on another platform as pQTL. The principle novelty in what is shown is their colocalisation with T1D risk variants and the MR analysis indicating a causal role in T1D. Other pQTL are effectively validation of reported associations in newborns or unvalidated new pQTL in a distinct and much smaller cohort. There is clear discussion of previous functional work on the 2 causal variants but no new data presented. The authors claim that their results may allow inclusion of protein quantification as biomarkers for risk stratification or for drug target identification but it isn't clear what this might add and no new data.

Methods

Use an established screening platform (GPPAD) to identify high risk children and an ongoing therapeutic trial of oral insulin nested within that platform.

Use dried blood spot from newborns to quantify 1985 proteins using OLink platform.

pQTL analysis on those proteins with colocalization and causal inference (mendelian randomisation) to investigate link to T1D. No discussion of why restricted to MR on *cis* only variants with *trans* variants excluded.

We thank the reviewer for their comment. Both *cis* and *trans* signals are now reported for both *coloc* and MR analysis.

Not clear how multiple correction was handled during the identification of 'significant' *coloc* for the 5 variants (and not explicit over how many variants tested from the meta-analysis).

Colocalization analysis was done using the R package Coloc, a well-established Bayesian method that uses the full summary statistics from the pQTL and GWAS studies, centered around the GWAS lead variant +/- 1Mb, to calculate the poster probability of the two signals sharing the same causal variant in the locus. This is also now stated in the main text. The strength of the method is that it evaluates all SNPs within the locus, rather than a single SNP, accounting for the linkage disequilibrium (LD) structure among SNPs. The number of SNPs used for each colocalization test is reported in the supplementary tables 5 and 6 (column “nvariants”) for *cis* and *trans* signals, respectively.

Correction of the multiple protein-phenotype colocalizations is a known analytical challenge with the analysis. Colocalization tests return posterior probabilities (PP) for different scenarios (e.g., PP for the presence of a shared causal variant, presence of separate causal variants, etc.) and it is not straightforward how this information could be incorporated in the prior probabilities of the Bayesian tests.

Recent large consortia studies, such as GTEx and Open Targets, have also not applied multiple-testing correction for all QTL-GWAS locus pairs in colocalization analyses.

For example, GTEx Consortium, did not perform multiple-testing correction in their colocalization study between eQTL data from 49 tissues and 87 GWAS. Similarly, Open Targets did not apply correction in their colocalization analyses between multiple QTL datasets and over 1,000 GWAS studies.

The GTEx Consortium, The GTEx Consortium atlas of genetic regulatory effects across human tissues. *Science* 369, 1318-1330(2020). <https://doi.org/10.1126/science.aaz1776>

Mountjoy, E., Schmidt, E.M., Carmona, M. et al. An open approach to systematically prioritize causal variants and genes at all published human GWAS trait-associated loci. *Nat Genet* 53, 1527–1533 (2021). <https://doi.org/10.1038/s41588-021-00945-5>

Results

Fig 1A is not a very helpful way to present the findings as much of what is shown is not clearly interpretable. Little actual data is presented - for example we are not shown the proteins tested, which overlap with the adult cohort and which are 'new'.

We have removed Figure 1A from the main manuscript since, in agreement with the reviewer, we believe the figure does not provide enough information to be included in the main text. We believe the figure still provides an interesting genome-wide overview of the results that it is not otherwise possible in 2D and we have therefore moved it to a new supplementary materials file. We have improved Figure 1B, which is now Figure 1, to show which results are novel and which are known. This information is also available in Supplementary Figure 3.

The colour legend for Fig1B appears to relate to cis/trans category only and not the magnitude of significance as presented.

The figure has been modified to clearly display the magnitude of effect. It now also displays novelty, as described in the previous comment.

It would also be relevant to show which pQTL that are reported elsewhere were not detected amongst the 467 proteins with a pQTL in newborns. It is claimed that pQTL are largely stable between newborns and adults, although we don't see what % of the 1985-467 = 1518 proteins that don't show a pQTL here, did show one in adults (I accept the power of detection is very different, but that is a relevant limitation to the discovery here).

The UKBB study had a much larger sample size and identified 2627 pQTL-targeted assays (as reported in Table 1 of Eldjarn et al.), which corresponds to 90% of the proteins analyzed

Eldjarn, G.H., Ferkingstad, E., Lund, S.H. et al. Large-scale plasma proteomics comparisons through genetics and disease associations. *Nature* 622, 348–358 (2023).
<https://doi.org/10.1038/s41586-023-06563-x>

We have now added an in-depth comparison to both the publicly available UKBB pQTL summary statistics and to a new analysis undertaken for the purposes of the revised manuscript, on a subset of UKBB participants with type 1 diabetes. We show that there is strong correlation ($R=0.77$; Supplementary Figure 4) between the effect sizes from our pQTL analysis and the full UKBB dataset. We also show replication of specific pQTLs in individuals with type 1 diabetes. We agree with the reviewer that the power difference between this study and the UKBB is substantial. This is reflected in the fact that 1329 (87.8%) of proteins that did not show a pQTL here, did show one in adults.

We aren't told how the 1985 proteins are selected (presumably based on availability through Olink panels, but those are more extensive than the proteins tested here).

The 1985 protein assays, out of 2941 initial assays included in the Olink panels, were selected based on passing stringent QC. The total number of measured proteins and the QC steps are now fully detailed in the methods.

The link between variants and detection is clearly an issue for all new and potentially for previously-reported pQTL also. Reader cannot easily define what is valid given the data presented. The issue potentially relates to any O-link quantified protein that is not validated with a secondary method, and that appears to include most of what is presented here.

We agree that affinity-based assay measurements could be affected by protein-structure-altering variants which would result in a *cis*-pQTL signal. For the proteins measured by both technologies, we have therefore now added information regarding correlation of protein level measurements between Olink and SomaScan from publicly available datasets. We have also queried eQTL databases to show that the *cis* variants are not only associated with protein levels but also with gene expression. For the majority of cases, we found good agreement between Olink and SomaScan measurements and all 4 novel *cis*-pQTLs have been detected as eQTLs in blood with the same direction of effect. While these additional analyses show that the reported results are robust, we agree that larger proteogenomic studies on other technologies such as mass-spec will be needed in the future for further validation.

Minimal discussion of the 'New' pQTL and why they are detected here and not elsewhere?

We have now performed a KEGG pathways over-representation analysis of the proteins targeted by novel pQTLs, and found a significant enrichment for the insulin signaling pathway. In the absence of a comparable newborn cohort, we have analyzed UKBB genotype and Olink data for a subset of participants with type 1 diabetes ($N=61$ adults). Although none of the novel *cis*-pQTLs reached significance in this smaller subgroup, we successfully replicated 3 *trans* signals related to T1D biology. SPTLC, a protein involved in the sphingolipid metabolism whose *cis*-pQTL was previously identified as associated with T-helper proportions in a T1D cohort

Chu X., Janssen WM. A., Koenen H., et al. A genome-wide functional genomics approach uncovers genetic determinants of immune phenotypes in type 1 diabetes *eLife* 11:e73709 (2022) <https://doi.org/10.7554/eLife.73709>

The IQ Motif Containing GTPase Activating Protein 2 (IQGAP2) was instead targeted by a SNP that sits 300kb from the insulin gene promoter. All of these new analyses and results are presented and discussed in the revised manuscript. the WAP, Kazal, immunoglobulin, Kunitz, and NTR domain-containing protein 2 (WFIKKN2), a loss-of-function variant affecting WFIKKN2 circulating protein levels has been associated with HOMA-IR levels

Ngo D, Benson MD, Long JZ, et al. Proteomic profiling reveals biomarkers and pathways in type 2 diabetes risk. *JCI Insight*. (2021) Mar 8;6(5):e144392
<https://doi.org/10.1172/jci.insight.144392>

No description of possible mechanism, although not likely to be simple given expression of ICA1 in pancreas and in Treg (with expression here presumably dictated more by the latter). ICA1 has been previously associated with T1D but its role in the disease is uncertain. As pointed out by the reviewer, it is expressed in both pancreas and Treg

Pesenacker A.M., Chen V., Gillies J. et al. Treg gene signatures predict and measure type 1 diabetes trajectory. *JCI Insight* (2019);4(6):e123879.
<https://doi.org/10.1172/jci.insight.123879>

Karges W., Pietropaolo M., Ackerley C.A. et al. Gene Expression of Islet Cell Antigen p69 in Human, Mouse, and Rat. *Diabetes* 1 April 1996; 45 (4): 513–521.
<https://doi.org/10.2337/diab.45.4.513>

both of which have a role in T1D. It is also observed in testis and there is a possibility that it has a neurotransmitter activity. Reference to its expression in the pancreas and its T1D-associated expression in Tregs has now been added to the discussion, but we have chosen not to speculate on mechanism.

Validation of most pQTL is not necessary (as they are previously reported and hence this study is validation, only in newborns). However, it would be necessary to validate the 'new' findings, both technically (with another platform) and in another cohort.

We thank the reviewer for this comment. As mentioned in the response to the comment further above, we have attempted replication of novel signals by accessing the full UKBB summary statistics to see if the novel results were also nominally significant in the UKBB. We have also extracted publicly available data on the correlation of Olink and SomaScan protein measurements and eQTL data from GTEx and eQTLGen to exclude that the novel results are technical and not biological. Finally, we gained access to UKBB genotype and Olink measurements in the subset of participants with type 1 diabetes. Altogether, we could replicate a large part of the findings and identified type 1 diabetes specific pQTLs. For those signals we could not replicate, new data for a proteogenomic study on a similar cohort, but with a different technology, will have to be generated. Given the scarcity of materials available from dried blood spots from newborns, such data generation on the same newborns is currently not possible.

Only 2 of the three cis pQTL run validated in Somalogic data and no attempt to validate the association with a secondary technology.

Olink and SomaScan are now the leading platforms for proteogenomic studies. Very scarce blood pQTL data are available from complementary technologies, such as mass-spec or ELISA. Our definition of novelty is based on comparison to an extensive database of 46 studies, 6 of which are from mass-spec measurements. Replication of the results using publicly available data is therefore not possible. We have now further attempted replication by accessing the full summary UKBB pQTL summary statistics and by generating new UKBB pQTL data on a subset of UKBB participants with T1D. Given the scarcity of materials available from dried blood spots from newborns, and the complexity of optimizing a new assay for dried blood spots, it was not possible to further replicate the results in newborns.

Trans-pQTL not run with mendelian randomisation and not made clear why the restriction to cis-variants was made.

Both cis and trans signals have now been included in the colocalization and Mendelian randomization analysis.

CTRB1 variant has several previously-described associations including pancreatic volume and T1D. The principal novel result here is the coloc and MR analysis suggesting causality. No new data on the putative mechanism is presented.

We agree that previous studies have found a correlation between the levels of the colocalizing proteins and T1D. However, these studies did not account for genetic factors, leaving the causal relationship between the protein levels and disease state unclear. Our findings demonstrate that genetic variants are linked to both protein levels and disease well before the onset of symptoms, at birth. This suggests that the direction of effect is SNP → protein → T1D, since a reverse temporal association is not possible. These results provide unique information that further strengthen previous links between these proteins and T1D. We agree that understanding the functional mechanism of the identified association is important but it is beyond the scope of the current study. We are currently planning to study the effect of CTRB1, as well as, IL7R and APOBR in a longitudinal setting once the clinical trial has ended.

IL7R - previous functional characterisation of the variant but nothing new presented here. Please see response to the comment above.

Central to the discussion is the proposal that new pQTL signals present at birth but not later could indicate an early pathology. This is very speculative, particularly as there is no validation of the 'new' pQTL signals, with no colocalisation or MR analysis of them.

With the new analyses, we have now shown that the new pQTLs are enriched for the insulin signaling pathway. We have also identified 3 novel pQTLs as possibly T1D specific, with replication in a small subset of UKBB, one of which has previously been reported as a T1D-specific QTL. The discussion has been extensively re-written and we hope the reviewer will agree with our interpretation of the results.

Similarly, it is claimed (in both abstract and discussion) that the proteins could be used as supportive biomarkers for early diagnosis, supporting GRS. However, GRS already have access to the SNP variants associated with T1D on which these pQTL are based and no data is presented to support their utility as biomarkers or therapeutic targets (although they could have been considered as either before this study given the previously demonstrated SNP association and, in some cases, demonstrated protein level association with disease).

We have removed reference to the use of the proteins as supportive biomarkers from the abstract since we have no data. However, we feel that it is warranted to briefly speculate that these have potential as predictive biomarkers with or without a polygenic risk score in the discussion. In combination with a PRS, there may be added value since although there is an association with the predisposing SNP, the values within genotypes of the SNP vary. As indicated, we do not have the data set to test whether the protein concentration is predictive for islet autoantibodies or T1D within genotypes. Furthermore, pQTLs only explain part of the protein heritability. Including the dynamic protein levels resulting from gene-environment interactions may complement the fixed genetic risk of the PRS model. To support this, there are several publications on prediction models based on UKBB Olink data showing increased accuracy of including protein levels for disease prediction

Carrasco-Zanini, J., Pietzner, M., Davitte, J. et al. Proteomic signatures improve risk prediction for common and rare diseases. *Nat Med* 30, 2489–2498 (2024).
<https://doi.org/10.1038/s41591-024-03142-z>

It has also been shown that protein-based prediction models can be more accurate than those based on PRS, but it is currently not known whether genetic-agnostic protein selection is better than genetically informed selection, or whether the two overlap. Whether the inclusion of the identified pQTLs or the protein levels in the GPPAD PRS model could help increase its accuracy will need to be addressed but it is beyond the scope of this manuscript.

REVIEWER COMMENTS

Reviewer #1 (Remarks to the Author):

The author responded all concerns
We thank the reviewer for their comment.

Reviewer #2 (Remarks to the Author):

The manuscript by authors has improved since the first version. I still have a few issues with some of the methodology and there are some outstanding queries after authors provided their previous response. It would be really helpful to have the changes made to the paper copied into the response to reviewers document, along with line numbers of where changes were implemented.

We thank the reviewer for their positive comment. We have ensured we quote changes to the paper along with line numbers in the response document.

For example, authors responded to some of my queries, but I still have a few questions:

In my previous comment I indicated that using $h4$ of > 0.6 defined as probable colocalization is not appropriate. I'm familiar with the papers the authors have shared and understand (and agree) with their sentiments, however I have two points: (1) categorising probabilities would be akin to categorisations of p-values where, for example, < 0.05 indicates evidence of an effect 0.05-0.1 indicates suggestive evidence, this should be avoided and instead the probabilities (as with p-values) should be interpreted as a continuum in light of the surrounding evidence; (2) the calculation of the probability of colocalization conditional on the presence of a causal variant is dependent upon p_{12} . As p_{12} changes so will this probability, therefore signals which are not robust may show larger probabilities simply because of p_{12} and window size. I am happy with the analyses but suggest the authors just rephrase such that the interpretation is given in light of other evidence and as a continuum, which is susceptible to priors and window size.

We thank the reviewer for their positive comment on our analyses and have now followed their recommendation to remove references to the categorization of PP4 in lines 191-196:

“We found evidence of a shared causal T1D risk variant with pQTLs for 5 proteins: CTRB1 (PP4 99.7%), CPA1 (PP4 99.7%; trans-pQTL), PNLIPRP1 (PP4 99.6%; trans-pQTL), APOBR (PP4 92.1%), and IL7R (PP4 68.8%) (Figure 2). The cis signal for CTRB1 and the trans signals for PNLIPRP1 and CPA1 shared the same lead variant, rs72802342, and the same colocalization signal with T1D.”

As a side note, the justification behind the choice of probably colocalization of $h4$ between 0.6 and 0.8 was that other papers had done the same. This is not a valid justification of why you selected certain methods - there are plenty of papers which repeat poor methods, which does not make it correct.

The reviewer's point is well-taken.

In response to my comment about the power calculation, authors again cited other papers that had used similar sample sizes. A more thorough approach is performing the power calculation. Just because there are smaller GWAS out there doesn't mean they are well performed studies.

We have now included a power calculation as Supplementary figure 5. The power curves show that our study had adequate power to detect associations within every minor allele frequency (MAF) bin. As reported in the previous response to reviewers, we have also overlayed our data to the estimated minimum detectable beta given our experimental design and MAFs (supplementary figure 3) and find that the curve of estimated beta coefficients for the 80% power curve closely follow the true

datapoints. These results suggest that, given the observed effect sizes, our study had adequate power to detect the reported pQTLs.

A methods section for the power curve generation has been added (lines 382-390):

“Power curves

The R package PowerEQTL was used to generate power curves as a function of minor allele frequency (MAF; supplementary figure 5). For the calculation, the sample size was set to 695 and the significance threshold to 5×10^{-8} . Since the protein levels were inverse normal transformed, the outcome standard deviation was set to 1. PowerEQTL was also used to estimate the minimum detectable beta for which we had a power of 50% and 80% as a function of MAF. The so calculated curves were then overlayed to the true data which closely followed the power lines. A data point that, for a given MAF, show an absolute beta estimate much smaller than the estimated minimum detectable beta from the power curves would be an indication of likely false positive.”

The Olink QC section has been updated following my previous comment. A description of the Olink technology would be useful here. Based on the updated reporting of whether proteins or samples were excluded - the defined criteria seem very strict (where samples with >10% warnings across assays were removed or proteins with >5% missingness across samples were removed). The latter removes 668 proteins, where there still may be some important information. I think it would be better to include more proteins here and to allow more missingness (for example, exclude proteins where there is >20% missingness for a protein). Including more proteins, but being fully transparent about missingness of all proteins and warnings may be a better approach. Or alternatively, using multiple imputation. Why were these criteria chosen?

A description of the Olink technology has now been added in lines 357-366.

“The Olink technology utilizes a Proximity Extension Assay in which a pair of protein-targeting antibodies are tagged with unique complementary oligonucleotide probes. Once bound to the target protein, the probes can hybridize to allow DNA amplification. The amplified signal is finally read using next-generation sequencing.

The samples of 770 children from the GPPAD POINT trial were processed by the Helmholtz Munich proteomics core facility. Reports were generated with data for 2941 proteins from the Olink Explore assay which comprises eight panels targeting inflammation (Inflammation I and II), oncology (Oncology I and II), cardiometabolic (Cardiometabolic I and II) and neurological (Neurology I and II) proteins.”

We agree with the reviewer that our Olink QC approach was particularly stringent and that we might have missed further true signals. Given the low starting material from dried blood spots, and the required adaptation of the Olink protocol, we decided to apply stringent QC criteria to minimize the chance of false positives.

Authors have attempted to add more detail to their Mendelian randomization analysis. However, the extra Mendelian randomization methodological detail is not as I would expect. The individual functions used in the TwoSampleMR package are not necessary information. As mentioned previously, giving a clear overview of the GWAS that performed was performed to give you your summary statistics etc are important. I would advise authors fill out and upload the STROBE-MR checklist to ensure they are following recommended guidelines. I would also advise that authors make their code for analyses available on GitHub or Zenodo so that analyses are reproducible.

We thank the reviewer for the clarification and we have now added information regarding the GWAS used for the outcome dataset. As suggested, we have filled in and provide the STROBE-MR checklist, as Supplementary Table 8.

The MR method section now reads (lines 470-481):

“Mendelian randomization

The R package TwoSampleMR v.0.5.647 was used to perform Mendelian randomization as an orthogonal validation step for the proteins whose cis-pQTL signals colocalized with T1D GWAS signals. Since a single cis-pQTL instrument was used for each of the proteins, the Wald ratio test was performed. For the outcome data, the utilised T1D GWAS study (accession number GCST90014023) is the result of inverse-variance weighted meta-analysis of 9 European cohorts. The meta-analysis has a combined sample size of 18,942 patients with T1D and 501,638 controls. The lead pQTL variants from the COJO conditional analysis (p -value $< 5 \times 10^{-8}$) were used as instrumental variables (Table 1). Betas and standard errors were available for all the instrumental variables from the T1D summary statistics (supplementary table 7). P-values were adjusted for multiple test correction using the Bonferroni method and a corrected p -value < 0.05 was considered significant. The STROBE-MR checklist is available as supplementary table 8.”

Some additional queries I have

- The "Comparison to adult proteogenomic data" section from line 534 is not written very clearly. Please ensure when writing methods that you are indicating why you have performed an analysis. For example, why were correlation coefficients used? I know it is to see whether they are measured similarly across technologies (after reading the response to reviewers), but this is not clear in the main manuscript in the methods.

We have now rephrased the text and have included the information about why we retrieved the Olink-SomaScan protein level correlation coefficients. The text now reads (lines 431-448):

“Comparison to adult proteogenomic data

Affinity-based assay measurements could be affected by protein-structure-altering variants which would result in a false cis-pQTL signal. To see whether the identified pQTL-targeted proteins are similarly detected across technologies, we retrieved the Olink-SomaScan protein level correlation coefficients from the Eldjarn, Ferkingstad and Lund et al. supplementary materials 4, which calculated the correlation of protein measurements between the two technologies from matching samples. For the 4, novel, cis-pQTL signals, we queried GTEx and eQTLGen to see if the pQTL SNPs have been previously detected as eQTL for the gene encoding the pQTL-targeted proteins with the same direction of effect following reference allele matching.

We also downloaded the full UKBB pQTL summary statistics, made available by Sun, Chiou, Traylor et al., through Synapse (project ID syn51364943) using the synapser R package. For each OlinkID-SNP pair, the chromosome, position, effect size, effect allele, minor allele frequency and p -value information were extracted. Pearson correlation was used to calculate the correlation coefficients between betas and MAFs in the UKBB and newborn datasets. We also used the full UKBB pQTL summary statistics to see if any previously unreported pQTL showed evidence of association in the UKBB while not reaching their study significance threshold.”

- Again, categorisation of results into $p < 0.05$ and $p > 0.05$ as significant and non-significant, respectively, should not be used. -Instead, strength of evidence should be reported.

We have now rephrased the main text to remove the categorization of p values. The text now reads (lines 163-167):

“We then queried the full UKBB-pQTL to see if the novel results in our study could be identified in the UKBB but did not reach their pQTL genome-wide threshold. We found that, out of 62 novel pQTLs, 16 showed some evidence of association in the UKBB with the same direction of effect. This included 1 of the 4 *cis* signals, targeting MRPL28, which had a p-value of 6.5×10^{-7} in the UKBB.”

-Olink having same protein on multiple assays is more of a technical control rather than validating the pQTL, as it is still performed in the same sample?

We agree with the reviewer that the 5 proteins measured on multiple assays should be treated as a technical control of the results. The results for these proteins have been reported with the intent of showing the technical reproducibility of the results. The text now reads (lines 104-107):

“Olink includes a set of 5 proteins measured on multiple assays, which can be used for quality control. For 2 out of 5 of these proteins (LMOD1 and IDO1), we identified *cis*-pQTL signals which were statistically significant for all three independent assay measurements that passed quality control, suggesting good reproducibility of the results.”

Reviewer #3 (Remarks to the Author):

Thank you to the authors for their thoughtful response to the queries raised. I appreciate their efforts to provide additional supportive data to bolster their conclusions and the additional interpretation of the results included in discussion.

We thank the reviewer for their positive comment.

For me, the biggest issue remains that the 'new' pQTLs presented are not validated, either in another cohort or with another technology (with significant known caveats with SNP variants influencing protein detection for the methods presented).

I fully appreciate the challenge of obtaining another cohort comparable to this, but the analyses presented do not show validation (with concordance of direction of effect not the same thing).

We appreciate the reviewer's point. In the absence of an independent newborn proteomic validation cohort, we have taken measures to ensure the robustness of our results. A query of eQTL databases showed that the *cis* variants are not only associated with protein but also with gene expression levels, suggesting that the identified *cis*-pQTLs are not likely to be the result of antibody-binding affinity. We also showed that 16 out of 62 novel pQTLs showed evidence of association in the UKBB, all with the same direction of effect. Nevertheless, we appreciate that it is not feasible to directly replicate the novel findings and have added this as a limitation of the study in the discussion. (lines 288-297):

“For the previously unreported signals, while we provide orthogonal evidence suggesting that many exhibit an association with the same direction of effect in publicly available eQTL and pQTL datasets, this does not constitute direct replication. The limited availability of dried blood spot samples from newborns, combined with the challenges of optimizing new assays for this material, made it infeasible to further replicate the findings in a newborn cohort. Additionally, the small sample size of the UKBB T1D cohort prevented us from determining whether the novel pQTLs are associated specifically with newborns, an increased risk of T1D in absence of the condition, or both. Future studies validating these novel findings in newborns with and without an elevated risk of T1D will be essential to fully understand the genetic regulation of these proteins.”

Response to comments in the pdf attachment

Responses to the first round of reviewers' comments are in italics. New reviewers' comments and new responses are denoted in purple.

Genetics of circulating proteins in newborn babies at high risk of type 1 diabetes

We thank the reviewers for their comments, which have helped improve the manuscript. We provide a point-by-point response below.

Reviewer 1

The present study reports a GWAS with almost 2,000 proteins measured in dried blood spots from 700 newborns that were at increased risk of developing T1D. The authors report 535 pQTLs, 62 of which have not been reported before. They show that pQTLs for CTRB1, APOBR, IL7R, CPA1, and PNLIPRP1 colocalize with T1D GWAS signals and implicate these five proteins in the aetiology of T1D using Mendelian Randomization. The paper is concise but to the point in that it reports novel pQTLs that are of clinical relevance.

I have only one major point to comment on, and that is the use of the term “causal” in relation to the MR analysis. Two sample MR using Wald ratios does not show causality on its own. The authors have not shown that all assumptions of MR are valid, i.e. it is known that the Olink platform may be prone to epitope changing variants. It is possible that genetic variants change the protein binding of the antibodies without affecting the protein levels. The computational result of the MR would be the same, but the conclusion that the levels of the affected proteins are causal for changes in the outcome is certainly not true in such cases. Also, proteins are measured in blood, whereas their effects on the outcomes may be through processes located in some organs. There is thus the possibility of horizontal pleiotropy between the same protein expressed in different tissues and their readout in blood that cannot be ruled out. It is even possible that a genetic variant increases protein levels in blood to compensate for lack of protein in some organs. In such cases the directionality of the MR estimate could be the inverse of the true effect and misguide clinical intervention.

We appreciate the comments regarding the potential influence of epitope-changing variants on the three cis-pQTL signals and the use of the term "causal". We agree that antibody affinity due to epitope changes could contribute to these signals. In order to address this concern, we have now queried the correlation between protein levels measured by two independent technologies, Olink and SomaScan, as reported in a recent study

Eldjarn, G.H., Ferkingstad, E., Lund, S.H. et al. Large-scale plasma proteomics comparisons through genetics and disease associations. Nature 622, 348–358 (2023). <https://doi.org/10.1038/s41586-023-06563-x>

We found that two out of the three cis-targeted proteins -CTRB1 and IL7R- were measured by both technologies. Both showed high correlation with Spearman coefficients of 0.73 and 0.78 for IL7R and CTRB1, respectively. In addition, the Olink pQTL signals for both CTRB1 and IL7R were also identified using SomaScan measurements. The high concordance between these two distinct measurement technologies strengthens the validity of our results and reduces the likelihood that the observed signals are solely artifacts of antibody binding issues. We have incorporated this new comparison into the revised manuscript. We also acknowledge that blood may not represent the primary tissue of effect for the identified pQTL signal colocalizations, particularly for CTRB1, which is a pancreatic enzyme, and that the directionality needs to be interpreted with caution. We have extended the discussion to include this among the limitations of the study and have rephrased the text throughout.

We also clarify in the text that the causality inferred from our MR analysis is statistical. While our findings provide strong evidence for an association, the biological causal role of these proteins in T1D pathogenesis requires further experimental validation.

Reviewer 2

Yiu Lin, Titino et al provide a study that performs a protein GWAS of Olink proteins in newborns at a higher risk for T1D. They identify both cis and trans pQTLs and attempt to put their findings into context with published GWAS of proteins in adults. Additional studies such as colocalization and Mendelian randomization have been performed in effects to explore whether the pQTLs may have an effect on T1D.

This is an interesting concept. However, I found the manuscript difficult to read, given the lack of headings of sections (abstract, intro, results, discussion). I also found the manuscript to be lacking detail, especially in the methods and results, therefore this manuscript needs more work to be publishable. There are some questions I have about some of the approaches used in the methods. I also think the interpretation of results may not be very accurate. Specific details are below.

We have extensively re-written and formatted the manuscript, which is now divided into sections with headings and sub-headings. We have also modified the text to include more information in all sections. We hope the new version of the manuscript is now easier to read.

Introduction

- Very brief introduction with 5 references. Authors need to give a more thorough overview of the current field.

Following the reformatting and extensive inclusion of detail, the introduction has also been extended.

Methods

Even though a reference to additional study information has been provided, factors such as the age of the children is important for the current study and therefore should be included in the manuscript. How do they define newborn?

For GPPAD participants included in this study, the median sampling age was 2 days and this information is now included in the manuscript.

If all are high risk of T1D, and no controls, will important signals for T1D be missed in the GWAS

The newborns enrolled in the GPPAD study have a >10% genetic risk of developing multiple beta-cell autoantibodies by the age of 6 years. The estimated prevalence of T1D in these high-risk infants was estimated to be 1.1%. Given the relatively low estimated prevalence, although much higher than the general population, we expect that T1D GWAS signals affecting the risk of developing the condition would remain detectable, albeit with likely reduced effect sizes.

Therefore, while it is possible that these signals did not reach genome-wide significance, we identified colocalization signals with T1D that were previously missed by Olink studies but identified in SomaScan pQTL studies, such as IL7R. On the other hand, pQTL signals specific for T1D would be more visible in this cohort. In support of this, we have added a new analysis in the revised manuscript that focuses on a subset of UKBB study participants with T1D, which allowed us to replicate three T1D-specific signals that were not significant in the full UKBB cohort.

Supp Table 2 lacks detail on each study and why they are being highlighted.

We have provided more information about the database in the manuscript and with reference to the original publication which contains the full description. We have also included more information about each study in Supplementary Table 2.

Line 222 - what does problematic proteins and problematic samples mean? What is this based on? Is it limit of detection?

We apologize for the vague description. We have now described in more detail the definitions for exclusion in the methods and have also added the number of excluded samples and proteins. Specifically, "problematic" was referring to the manufacturer's assay and sample QC warnings.

Line 262, $h4$ of > 0.6 defined as probable colocalization. This may be a bit too lenient.

The $H4$ threshold for colocalization can vary across studies, with recent examples using thresholds as low as 0.5

Gudjonsson, A., Gudmundsdottir, V., Axelsson, G.T. et al. A genome-wide association study of serum proteins reveals shared loci with common diseases. Nat Commun 13, 480 (2022). <https://doi.org/10.1038/s41467-021-27850-z>

Oliva, M., Demanelis, K., Lu, Y. et al. DNA methylation QTL mapping across diverse human tissues provides molecular links between genetic variation and complex traits. Nat Genet 55, 112–122 (2023). <https://doi.org/10.1038/s41588-022-01248-z>

Further review of $IL7R$ colocalization results shows that $H4/(H4 + H3)$, which represents the probability of colocalization conditional on the presence of a causal variant, as described by

Zuber, V., Grinberg, N.F., Gill, D. et al. Combining evidence from Mendelian randomization and colocalization: Review and comparison of approaches. The American Journal of Human Genetics, 109, 5 (2022). <https://doi.org/10.1016/j.ajhg.2022.04.001>

is 0.68 and that $H4/H3 > 2$, indicating that the probability of the two traits sharing the same causal variant is twice as high as the probability of the two traits having two independent causal variants. However, we acknowledge that we cannot fully exclude the latter given that $H3$ is non-zero ($H3 = 0.16$). We believe these results are still noteworthy, and we have revised the manuscript to categorize the results as having high support ($H4 > 0.8$) and medium support ($H4 > 0.6$).

From line 267 - description of Mendelian randomization far too brief. Which GWAS were used for exposure and outcome? What p value for SNPs?

The methods section of the MR analysis has now been extended with substantially higher level of detail.

Did authors perform any power calculations for this GWAS? Small sample size used.

We agree with the reviewer that, compared to biobank-scale projects, the sample size is modest. However, it is larger or only moderately smaller than other recent pQTL studies such as

Gilly, A., Park, YC., Png, G. et al. Whole-genome sequencing analysis of the cardiometabolic proteome. Nat Commun 11, 6336 (2020). <https://doi.org/10.1038/s41467-020-20079-2> – sample size = 1328

Koprulu, M., Carrasco-Zanini, J., Wheeler, E. et al. Proteogenomic links to human metabolic diseases. Nat Metab 5, 516–528 (2023). <https://doi.org/10.1038/s42255-023-00753-7> – sample size = 1180

Steinberg, J., Southam, L., Roumeliotis, T.I. et al. A molecular quantitative trait locus map for osteoarthritis. Nat Commun 12, 1309 (2021). <https://doi.org/10.1038/s41467-021-21593-7> – sample size = 115

Among these, Koprulu et al. already demonstrated that smaller cohorts can still contribute to the identification of new biologically relevant proteogenomic links to disease. We hope that the reviewer will agree with us that this is the first proteogenomic study from newborn dried blood spots (median of 2 days) and, therefore, not comparable to previous studies on easily accessible tissues (blood) from adults.

We have also added a deep comparison of our results to the extended UKBB pQTL data and have also generated new data on a subset of UKBB participants with T1D. We have shown that one of our novel cis-pQTL signals is also nominally significant in the full UKBB cohort but did not reach genome-wide significance (p -value = 6.5×10^{-7}) and that 3 signals are specific to the T1D-UKBB subgroup.

Therefore, despite the smaller sample size, our findings provide novel insights on the genetic regulation of protein levels in this valuable newborn cohort. To further address power concerns, we have also generated power curves and overlaid them to our results (supplementary figure 3) demonstrating that, given the observed effect sizes, we have sufficient power to detect the reported novel pQTLs.

Figure 1 - key for p value threshold unclear, all points for p value thresholds look the same. Points overlaying which changes the transparency of points. Maybe use size of the points for p value scale. *The new figure included in the manuscript now uses the point size instead of transparency. We believe the suggestion improved the readability of the figure and we thank the reviewer for the suggestion.*

When referencing Figure 1 - could make more specific references to Figure 1A/1B.

Following the reformatting of the manuscript, Figure 1A has been moved to the supplementary materials and referencing Figure 1B is no longer needed.

Figure 2 - Axis labelled protein is confusing, how are these sorted?
The proteins used in Figure 1A are alphabetically sorted. This is now explained in the figure caption.

Authors could provide more of a summary of the pQTLs identified more broadly.

We have included more general information about the identified pQTLs, such as the number of cis-pQTLs regulating protein levels in trans and distance of lead cis-pQTLs from their gene target transcription start site.

They mentioned that one SNP is cis for one protein and trans for another. What about all the other pQTLs from the GWAS? How many cis pQTLs were also trans pQTLs for other proteins?
As also suggested in the comment above, we have now added an extended summary of the pQTL results, which also includes the number of cis signals which were also trans for other proteins.

From line 118 - if the SNP for CTRB1 also associates with other proteins, does this suggest that there may be pleiotropy? Is the MR estimate truly estimating an effect of CTRB1 on T1D?

The regulatory mechanisms of trans-pQTL signals are not known. For eQTLs, which we might assume follow the same mechanisms, they are thought to be at least partially mediated through cis effects

Yang, F., Wang, J., The GTEx Consortium et al. Identifying cis-mediators for trans-eQTLs across many human tissues using genomic mediation analysis. *Genome Res.* 27, 1859-1871 (2017).
<https://doi.org/10.1101/gr.216754.116>

If that is the case for this locus, the trans-pQTL signals would be mediated through CTRB1 cis-pQTL. The observation that the cis and trans colocating signals share that same set of SNPs would indeed support this assumption but future experimental studies will be needed to confirm it.

Unclear where the discussion starts.

The whole manuscript has been re-formatted, and a clear Discussion section is now included.

Limitations need to be addressed.

The Discussion now includes a paragraph detailing the limitations of the study.

Sentence on 149 does not seem justified - you have identified pQTLs within a high-risk group, but have not shown that the levels of this protein are different compared to controls.

We agree with the reviewer and the sentence has now been modified:

"Whether the novel results that could not be identified by biobank-scale studies in adults, are due to increased risk of T1D in absence of the condition, or to newborn age, or a combination of the two, will require future replication."

No conflict of interest or author contributions

We have now added the conflict of interest and author contributions statements to the manuscript.

Reviewer 3

The authors construct a genome-wide pQTL map of 1985 proteins in 695 newborns at high risk of T1D, showing 535pQTLs (66%cis) in 437 unique proteins of which 62 are novel. Five pQTL, selected from overlap with T1D associated variants, show colocalisation with T1D susceptibility loci with Mendelian Randomisation implicating them in the aetiology of T1D.

The novel variants identified are difficult to interpret given the technical concerns over the platform (affinity impacted by protein-coding variants) and a lack of validation. Most pQTL identified are previously reported and two of the three 'causal' variants identified (IL7R, CTRB1) have supportive evidence on another platform as pQTL. The principle novelty in what is shown is their colocalisation with T1D risk variants and the MR analysis indicating a causal role in T1D. Other pQTL are effectively validation of reported associations in newborns or unvalidated new pQTL in a distinct and much smaller cohort. There is clear discussion of previous functional work on the 2 causal variants but no new data presented. The authors claim that their results may allow inclusion of protein quantification as biomarkers for risk stratification or for drug target identification but it isn't clear what this might add and no new data.

Methods

Use an established screening platform (GPPAD) to identify high risk children and an ongoing therapeutic trial of oral insulin nested within that platform.

Use dried blood spot from newborns to quantify 1985 proteins using OLink platform. pQTL analysis on those proteins with colocalization and causal inference (mendelian randomisation) to investigate link to T1D. No discussion of why restricted to MR on cis only variants with trans variants excluded.

We thank the reviewer for their comment. Both cis and trans signals are now reported for both coloc and MR analysis.

Not clear how multiple correction was handled during the identification of 'significant' coloc for the 5 variants (and not explicit over how many variants tested from the meta-analysis).

Colocalization analysis was done using the R package Coloc, a well-established Bayesian method that uses the full summary statistics from the pQTL and GWAS studies, centered around the GWAS lead variant +/- 1Mb, to calculate the poster probability of the two signals sharing the same causal variant in the locus. This is also now stated in the main text. The strength of the method is that it evaluates all SNPs within the locus, rather than a single SNP, accounting for the linkage disequilibrium (LD) structure among SNPs. The number of SNPs used for each colocalization test is reported in the supplementary tables 5 and 6 (column "nvariants") for cis and trans signals, respectively.

Correction of the multiple protein-phenotype colocalizations is a known analytical challenge with the analysis. Colocalization tests return posterior probabilities (PP) for different scenarios (e.g., PP for the presence of a shared causal variant, presence of separate causal variants, etc.) and it is not straightforward how this information could be incorporated in the prior probabilities of the Bayesian tests.

Recent large consortia studies, such as GTEx and Open Targets, have also not applied multiple-testing correction for all QTL-GWAS locus pairs in colocalization analyses. For example, GTEx Consortium, did not perform multiple-testing correction in their colocalization study between eQTL data from 49 tissues and 87 GWAS. Similarly, Open Targets did not apply correction in their colocalization analyses between multiple QTL datasets and over 1,000 GWAS studies.

The GTEx Consortium, The GTEx Consortium atlas of genetic regulatory effects across human tissues. Science 369, 1318-1330(2020). <https://doi.org/10.1126/science.aaz1776>

Mountjoy, E., Schmidt, E.M., Carmona, M. et al. An open approach to systematically prioritize causal variants and genes at all published human GWAS trait-associated loci. Nat Genet 53, 1527–1533 (2021). <https://doi.org/10.1038/s41588-021-00945-5>

Results

Fig 1A is not a very helpful way to present the findings as much of what is shown is not clearly interpretable. Little actual data is presented - for example we are not shown the proteins tested, which overlap with the adult cohort and which are 'new'.

We have removed Figure 1A from the main manuscript since, in agreement with the reviewer, we believe the figure does not provide enough information to be included in the main text. We believe the figure still provides an interesting genome-wide overview of the results that it is not otherwise possible in 2D and we have therefore moved it to a new supplementary materials file. We have improved Figure 1B, which is now Figure 1, to show which results are novel and which are known. This information is also available in Supplementary Figure 3.

The colour legend for Fig1B appears to relate to cis/trans category only and not the magnitude of significance as presented.

The figure has been modified to clearly display the magnitude of effect. It now also displays novelty, as described in the previous comment.

It would also be relevant to show which pQTL that are reported elsewhere were not detected amongst the 467 proteins with a pQTL in newborns. It is claimed that pQTL are largely stable between newborns and adults, although we don't see what % of the 1985-467 = 1518 proteins that don't show a pQTL here, did show one in adults (I accept the power of detection is very different, but that is a relevant limitation to the discovery here).

The UKBB study had a much larger sample size and identified 2627 pQTL-targeted assays (as reported in Table 1 of Eldjarn et al.), which corresponds to 90% of the proteins analyzed

Eldjarn, G.H., Ferkingstad, E., Lund, S.H. et al. Large-scale plasma proteomics comparisons through genetics and disease associations. Nature 622, 348–358 (2023). <https://doi.org/10.1038/s41586-023-06563-x>

We have now added an in-depth comparison to both the publicly available UKBB pQTL summary statistics and to a new analysis undertaken for the purposes of the revised manuscript, on a subset of UKBB participants with type 1 diabetes. We show that there is strong correlation ($R=0.77$; Supplementary Figure 4) between the effect sizes from our pQTL analysis and the full UKBB dataset. We also show replication of specific pQTLs in individuals with type 1 diabetes. We agree with the reviewer that the power difference between this study and the UKBB is substantial. This is reflected in the fact that 1329 (87.8%) of proteins that did not show a pQTL here, did show one in adults.

This power difference remains a major obstacle for the study's design and is reflected in this result. This has large implications for interpretation of the results presented. Underpowered analysis can generate false positives as well as miss true positives. While the concordant direction of effect in the larger cohort is supportive, it isn't validation (I presume these are not similarly significant in the other cohort otherwise this would be stated).

We agree with the reviewer that a concordant direction of effect is supportive but it is not a direct validation. To address the concerns regarding power, we have now generated power curves as a function of minor allele frequency (Supplementary Table 5). We show that our study had adequate power for every minor allele frequency bin. We further estimated the minimum beta that could be detected with 80% and 50% power as a function of minor allele frequency, given our study design. These have then been overlaid to the true data, showing that we had enough power to detect the novel pQTL given the observed betas.

We aren't told how the 1985 proteins are selected (presumably based on availability through Olink panels, but those are more extensive than the proteins tested here).

The 1985 protein assays, out of 2941 initial assays included in the Olink panels, were selected based on passing stringent QC. The total number of measured proteins and the QC steps are now fully detailed in the methods.

The link between variants and detection is clearly an issue for all new and potentially for previously-reported pQTL also. Reader cannot easily define what is valid given the data presented. The issue potentially relates to any O-link quantified protein that is not validated with a secondary method, and that appears to include most of what is presented here.

We agree that affinity-based assay measurements could be affected by protein-structure-altering variants which would result in a cis-pQTL signal. For the proteins measured by both technologies, we have therefore now added information regarding correlation of protein level measurements between Olink and SomaScan from publicly available datasets. We have also queried eQTL databases to show that the cis variants are not only associated with protein levels but also with gene expression. For the majority of cases, we found good agreement between Olink and SomaScan measurements and all 4 novel cis-pQTLs have been detected as eQTLs in blood with the same direction of effect. While these additional analyses show that the reported results are robust, we agree that larger proteogenomic studies on other technologies such as mass-spec will be needed in the future for further validation.

This is a major caveat and limits confidence in any result not validated with a second method. This affects much of what is shown. I don't see why this is needed in the future, and not for the work presented here?

Gene expression levels measured by RNA sequencing would not be affected by missense variants disrupting the protein aptamer binding site (Olink assay) and the eQTLs would not be expected to share the same directionality if the pQTL signal was due to antibody affinity. Further, we queried the GTEx and eQTLGen eQTL databases and found that all 4 novel cis-pQTLs have been previously detected as eQTLs in blood with the same direction of effect, further suggesting that the result is not driven by antibody affinity.

Minimal discussion of the 'New' pQTL and why they are detected here and not elsewhere?

We have now performed a KEGG pathways over-representation analysis of the proteins targeted by novel pQTLs, and found a significant enrichment for the insulin signaling pathway. In the absence of a comparable newborn cohort, we have analyzed UKBB genotype and Olink data for a subset of participants with type 1 diabetes (N=61 adults). Although none of the novel cis-pQTLs reached significance in this smaller subgroup, we successfully replicated 3 trans signals related to T1D biology. SPTLC, a protein involved in the sphingolipid metabolism whose cis-pQTL was previously identified as associated with T-helper proportions in a T1D cohort

I accept the challenges of obtaining a replication cohort here, but the risk of false discovery is high and the above is a failure of validation. The majority of observed associations don't validate in the larger public cohort either, highlighting the risk of false positive associations (or, you could argue, associations that are unique to neonates vs adults).

We would like to highlight that only 62 out of 535 pQTLs did not validate in the larger cohort. Of the ones that did not validate, meaning that they did not reach genome-wide significance, 16 still showed some evidence of association in the larger cohort. As also suggested by the reviewer, these previously unreported signals might be unique to neonates or to neonates at increased risk of T1D. The negative findings in the UKBB T1D cohort might suggest the former but the very small sample size of the UKBB T1D cohort severely limits the conclusions and, as stated in our previous response, the proteins targeted by previously unreported pQTLs are enriched for the insulin signaling pathway. The point of validation has been included as a limitation in the discussion. (lines 288-297):

“For the previously unreported signals, while we provide orthogonal evidence suggesting that many exhibit an association with the same direction of effect in publicly available eQTL and pQTL datasets, this does not constitute direct replication. The limited availability of dried blood spot samples from newborns, combined with the challenges of optimizing new assays for this material, made it infeasible to further replicate the findings in a newborn cohort. Additionally, the small sample size of the UKBB T1D cohort prevented us from determining whether the novel pQTLs are associated

specifically with newborns, an increased risk of T1D in absence of the condition, or both. Future studies validating these novel findings in newborns with and without an elevated risk of T1D will be essential to fully understand the genetic regulation of these proteins.”

Chu X., Janssen WM. A., Koenen H., et al. A genome-wide functional genomics approach uncovers genetic determinants of immune phenotypes in type 1 diabetes eLife 11:e73709 (2022) <https://doi.org/10.7554/eLife.73709>

The IQ Motif Containing GTPase Activating Protein 2 (IQGAP2) was instead targeted by a SNP that sits 300kb from the insulin gene promoter. All of these new analyses and results are presented and discussed in the revised manuscript. the WAP, Kazal, immunoglobulin, Kunitz, and NTR domain-containing protein 2 (WFIKKN2), a loss-of-function variant affecting WFIKKN2 circulating protein levels has been associated with HOMA-IR levels

Ngo D, Benson MD, Long JZ, et al. Proteomic profiling reveals biomarkers and pathways in type 2 diabetes risk. JCI Insight. (2021) Mar 8;6(5):e144392 <https://doi.org/10.1172/jci.insight.144392>

No description of possible mechanism, although not likely to be simple given expression of ICA1 in pancreas and in Treg (with expression here presumably dictated more by the latter). *ICA1 has been previously associated with T1D but its role in the disease is uncertain. As pointed out by the reviewer, it is expressed in both pancreas and Treg*

Pesenacker A.M., Chen V., Gillies J. et al. Treg gene signatures predict and measure type 1 diabetes trajectory. JCI Insight (2019);4(6):e123879. <https://doi.org/10.1172/jci.insight.123879>

Karges W., Pietropaolo M., Ackerley C.A. et al. Gene Expression of Islet Cell Antigen p69 in Human, Mouse, and Rat. Diabetes 1 April 1996; 45 (4): 513–521. <https://doi.org/10.2337/diab.45.4.513>

both of which have a role in T1D. It is also observed in testis and there is a possibility that it has a neurotransmitter activity. Reference to its expression in the pancreas and its T1D-associated expression in Tregs has now been added to the discussion, but we have chosen not to speculate on mechanism.

Validation of most pQTL is not necessary (as they are previously reported and hence this study is validation, only in newborns). However, it would be necessary to validate the 'new' findings, both technically (with another platform) and in another cohort.

We thank the reviewer for this comment. As mentioned in the response to the comment further above, we have attempted replication of novel signals by accessing the full UKBB summary statistics to see if the novel results were also nominally significant in the UKBB. We have also extracted publicly available data on the correlation of Olink and SomaScan protein measurements and eQTL data from GTEx and eQTLGen to exclude that the novel results are technical and not biological. Finally, we gained access to UKBB genotype and Olink measurements in the subset of participants with type 1 diabetes. Altogether, we could replicate a large part of the findings and identified type 1 diabetes specific pQTLs. For those signals we could not replicate, new data for a proteogenomic study on a similar cohort, but with a different technology, will have to be generated. Given the scarcity of materials available from dried blood spots from newborns, such data generation on the same newborns is currently not possible.

“Altogether, we could replicate a large part of the findings and identified type 1 diabetes specific pQTLs”

I don't think this statement is supported by what is presented. The key new pQTL didn't validate with concordant effect direction being a much weaker signal.

We have now rephrased the text as follows (lines 288-297):

“For the previously unreported signals, while we provide orthogonal evidence suggesting that many exhibit an association with the same direction of effect in publicly available eQTL and pQTL datasets, this does not constitute direct replication. The limited availability of dried blood spot samples from newborns, combined with the challenges of optimizing new assays for this material, made it infeasible to further replicate the findings in a newborn cohort. Additionally, the small sample size of the UKBB T1D cohort prevented us from determining whether the novel pQTLs are associated specifically with newborns, an increased risk of T1D in absence of the condition, or both. Future studies validating these novel findings in newborns with and without an elevated risk of T1D will be essential to fully understand the genetic regulation of these proteins.”

Only 2 of the three cis pQTL run validated in Somalogic data and no attempt to validate the association with a secondary technology.

Olink and SomaScan are now the leading platforms for proteogenomic studies. Very scarce blood pQTL data are available from complementary technologies, such as mass-spec or ELISA. Our definition of novelty is based on comparison to an extensive database of 46 studies, 6 of which are from mass-spec measurements. Replication of the results using publicly available data is therefore not possible. We have now further attempted replication by accessing the full summary UKBB pQTL summary statistics and by generating new UKBB pQTL data on a subset of UKBB participants with T1D. Given the scarcity of materials available from dried blood spots from newborns, and the complexity of optimizing a new assay for dried blood spots, it was not possible to further replicate the results in newborns.

This statement is a more accurate caveat to the data presented, although unavoidably limits the interest and confidence in the findings.

The limitation has been addressed in the discussion (lines 288-297):

“For the previously unreported signals, while we provide orthogonal evidence suggesting that many exhibit an association with the same direction of effect in publicly available eQTL and pQTL datasets, this does not constitute direct replication. The limited availability of dried blood spot samples from newborns, combined with the challenges of optimizing new assays for this material, made it infeasible to further replicate the findings in a newborn cohort. Additionally, the small sample size of the UKBB T1D cohort prevented us from determining whether the novel pQTLs are associated specifically with newborns, an increased risk of T1D in absence of the condition, or both. Future studies validating these novel findings in newborns with and without an elevated risk of T1D will be essential to fully understand the genetic regulation of these proteins.”

Trans-pQTL not run with mendelian randomisation and not made clear why the restriction to cis-variants was made.

Both cis and trans signals have now been included in the colocalization and Mendelian randomization analysis.

CTRB1 variant has several previously-described associations including pancreatic volume and T1D. The principal novel result here is the coloc and MR analysis suggesting causality. No new data on the putative mechanism is presented.

We agree that previous studies have found a correlation between the levels of the colocalizing proteins and T1D. However, these studies did not account for genetic factors, leaving the causal

relationship between the protein levels and disease state unclear. Our findings demonstrate that genetic variants are linked to both protein levels and disease well before the onset of symptoms, at birth. This suggests that the direction of effect is SNP → protein → T1D, since a reverse temporal association is not possible. These results provide unique information that further strengthen previous links between these proteins and T1D. We agree that understanding the functional mechanism of the identified association is important but it is beyond the scope of the current study. We are currently planning to study the effect of CTRB1, as well as, IL7R and APOBR in a longitudinal setting once the clinical trial has ended.

Thanks – this is a fair response and a good point.

IL7R - previous functional characterisation of the variant but nothing new presented here.

Please see response to the comment above.

Central to the discussion is the proposal that new pQTL signals present at birth but not later could indicate an early pathology. This is very speculative, particularly as there is no validation of the 'new' pQTL signals, with no colocalisation or MR analysis of them.

With the new analyses, we have now shown that the new pQTLs are enriched for the insulin signaling pathway. We have also identified 3 novel pQTLs as possibly T1D specific, with replication in a small subset of UKBB, one of which has previously been reported as a T1D-specific QTL. The discussion has been extensively re-written and we hope the reviewer will agree with our interpretation of the results.

Similarly, it is claimed (in both abstract and discussion) that the proteins could be used as supportive biomarkers for early diagnosis, supporting GRS. However, GRS already have access to the SNP variants associated with T1D on which these pQTL are based and no data is presented to support their utility as biomarkers or therapeutic targets (although they could have been considered as either before this study given the previously demonstrated SNP association and, in some cases, demonstrated protein level association with disease).

We have removed reference to the use of the proteins as supportive biomarkers from the abstract since we have no data. However, we feel that it is warranted to briefly speculate that these have potential as predictive biomarkers with or without a polygenic risk score in the discussion. In combination with a PRS, there may be added value since although there is an association with the predisposing SNP, the values within genotypes of the SNP vary. As indicated, we do not have the data set to test whether the protein concentration is predictive for islet autoantibodies or T1D within genotypes. Furthermore, pQTLs only explain part of the protein heritability. Including the dynamic protein levels resulting from gene-environment interactions may complement the fixed genetic risk of the PRS model. To support this, there are several publications on prediction models based on UKBB Olink data showing increased accuracy of including protein levels for disease prediction

Carrasco-Zanini, J., Pietzner, M., Davitte, J. et al. Proteomic signatures improve risk prediction for common and rare diseases. Nat Med 30, 2489–2498 (2024). <https://doi.org/10.1038/s41591-024-03142-z>

It has also been shown that protein-based prediction models can be more accurate than those based on PRS, but it is currently not known whether genetic-agnostic protein selection is better than genetically informed selection, or whether the two overlap. Whether the inclusion of the identified pQTLs or the protein levels in the GPPAD PRS model could help increase its accuracy will need to be addressed but it is beyond the scope of this manuscript.

REVIEWER COMMENTS

Reviewer #3 (Remarks to the Author):

The authors have conceded that they do not replicate the new findings presented and this has now been acknowledged in the manuscript.

We thank the reviewer for their comments and feedback on the manuscript.

Reviewer #4 (Remarks to the Author):

I have only reviewed the MR section of the manuscript as per my expertise.

We thank the reviewer for their comments.

This MR appears to be appropriately conducted however there are some details lacking and the results are not surprising given how the results were selected to be included in the MR.

We appreciate the acknowledgment that the Mendelian randomization (MR) analysis was appropriately conducted. We have included the additional details about the MR analysis as described below.

Regarding the details of the MR the authors should be explicit that they use only a single SNP for each MR (and not multiple SNPs from the same region). The strength of the association between the SNP and the exposure should be calculated and stated using an F-statistic.

The information that a single SNP was used for each MR test had previously only been included in the methods section of the MR analysis. To increase clarity, we have now also specified this in the main text, which now reads at page 5 lines 207-210:

“The top independent pQTL for both cis- and trans-pQTL SNPs identified by GCTA-COJO were therefore used as instrumental variables - for each protein, only the lead SNP was used as instrumental variable.”

We have also now calculated the F-statistic for the SNP-exposure association and have added the information to the supplementary table ST7_T1D_MR_Ivs. We have also added further information regarding the instrumental variables used in the MR analysis such as effect allele, allele frequency, beta and standard error. This information was previously only present in the summary of pQTL results within the supplementary table 1 but it is now also included in the supplementary table ST7_T1D_MR_Ivs for clarity. The following statements have also been included in the main results at page 5 lines 210-211:

“The strength of the association between the IVs and the exposure was further assessed with the F-statistic. All the IVs had an F-statistic > 15 and were retained in the analysis.”

And in the methods section page 10 lines 491-493:

“The F-statistic, defined as β^2/se^2 , was used to determine the strength of the association between IVs and the exposure. All the tested IVs had an F-statistic > 15 (supplementary table 7).”

My second comment regards more the use of MR in this setting and how these exposures were selected to be included in the MR. MR is most definitely not an orthogonal test for causality to the colocalisation that was performed in the previous step (as is currently stated in the methods). Both analyses are using the same data to test the correlation between the association of a SNP with the exposure and T1D, but in different ways. As only exposures for which the coloc showed strong evidence of colocalization with T1D it is unsurprising that they all show strong effects in the MR as they have only selected the exposures where we would expect to see an effect. The language around the interpretation of these results in the paper currently is too strong for what the results actually show.

We agree with the reviewer that the MR analysis cannot be considered orthogonal to colocalization since they both used the same data. We have now replaced the word “orthogonal” with “complementary”.

We also modified the last paragraph of the results as follows (page 5 lines 214-218):

“The colocalization and MR results suggest a potential link between circulating protein levels at birth and the development of T1D later in life (Table 1). However, it is important to note that the MR analysis relied on a single instrumental variable, which limits the robustness of the causal inference due to the inability to fully test for pleiotropy or validate the assumptions underlying the analysis.”

A limitation of all MR studies is that the assumption of no pleiotropic effects of the SNPs on the outcome. In this case it is not possible to test this assumption and so no further analysis is needed. However the limits on the ability of MR to test for causality and the assumptions required should be stated in the final paragraph of the results.

As also reported above, we have now rephrased the last paragraph of the results to state the limitations of the MR analysis using a single instrumental variable (page 5 lines 214-218):

“The colocalization and MR results suggest a potential link between circulating protein levels at birth and the development of T1D later in life (Table 1). However, it is important to note that the MR analysis relied on a single instrumental variable, which limits the robustness of the causal inference due to the inability to fully test for pleiotropy or validate the assumptions underlying the analysis.”

Finally, how should the beta’s from the MR be interpreted? What scale are these results on? Are they log odds ratios? If so the odds ratios should also be reported to enable interpretation of the potential magnitude of the effect estimated by the MR.

The effect size for the outcome had been reported in the original publication as the log-odds ratio from inverse-weighted meta-analysis which represents the unit increase or decrease risk of having T1D. The effect size for the exposure is the beta of z-score transformed Olink NPX values. The effect size of the Wald ratio test, therefore, represents the T1D risk per one standard deviation of the genetically predicted protein levels. This information is now included in the Table 1 legend and in the Methods page 10 lines 493-495:

“The effect size of the Wald ratio test represents the T1D risk per one standard deviation of the genetically predicted protein levels”

Following the reviewer request, the odds ratios of the outcome have also been added to the supplementary table ST7_T1D_MR_lvs.

Genetics of circulating proteins in newborn babies at high risk of type 1 diabetes

We thank the reviewers for their comments, which have helped improve the manuscript. We provide a point-by-point response below.

Reviewer 1

The present study reports a GWAS with almost 2,000 proteins measured in dried blood spots from 700 newborns that were at increased risk of developing T1D. The authors report 535 pQTLs, 62 of which have not been reported before. They show that pQTLs for CTRB1, APOBR, IL7R, CPA1, and PNLIPRP1 colocalize with T1D GWAS signals and implicate these five proteins in the aetiology of T1D using Mendelian Randomization. The paper is concise but to the point in that it reports novel pQTLs that are of clinical relevance.

I have only one major point to comment on, and that is the use of the term “causal” in relation to the MR analysis. Two sample MR using Wald ratios does not show causality on its own. The authors have not shown that all assumptions of MR are valid, i.e. it is known that the Olink platform may be prone to epitope changing variants. It is possible that genetic variants change the protein binding of the antibodies without affecting the protein levels. The computational result of the MR would be the same, but the conclusion that the levels of the affected proteins are causal for changes in the outcome is certainly not true in such cases. Also, proteins are measured in blood, whereas their effects on the outcomes may be through processes located in some organs. There is thus the possibility of horizontal pleiotropy between the same protein expressed in different tissues and their readout in blood that cannot be ruled out. It is even possible that a genetic variant increases protein levels in blood to compensate for lack of protein in some organs. In such cases the directionality of the MR estimate could be the inverse of the true effect and misguide clinical intervention.

We appreciate the comments regarding the potential influence of epitope-changing variants on the three cis-pQTL signals and the use of the term "causal". We agree that antibody affinity due to epitope changes could contribute to these signals. In order to address this concern, we have now queried the correlation between protein levels measured by two independent technologies, Olink and SomaScan, as reported in a recent study

Eldjarn, G.H., Ferkingstad, E., Lund, S.H. et al. Large-scale plasma proteomics comparisons through genetics and disease associations. *Nature* 622, 348–358 (2023).
<https://doi.org/10.1038/s41586-023-06563-x>

We found that two out of the three cis-targeted proteins -CTRB1 and IL7R- were measured by both technologies. Both showed high correlation with Spearman coefficients of 0.73 and 0.78 for IL7R and CTRB1, respectively. In addition, the Olink pQTL signals for both CTRB1 and IL7R were also identified using SomaScan measurements. The high concordance between these two distinct measurement technologies strengthens the validity of our results and reduces the likelihood that the observed signals are solely artifacts of antibody binding issues. We have incorporated this new comparison into the revised manuscript. We also acknowledge that blood may not represent the primary tissue of effect for the identified pQTL signal colocalizations, particularly for CTRB1, which is a pancreatic enzyme, and that the directionality needs to be interpreted with caution. We have extended the discussion to include this among the limitations of the study and have rephrased the text throughout. We also clarify in the text that the causality inferred from our MR analysis is statistical. While our findings provide strong evidence for an association, the biological causal role of these proteins in T1D pathogenesis requires further experimental validation.

Reviewer 2

Yiu Lin, Titino et al provide a study that performs a protein GWAS of Olink proteins in newborns at a higher risk for T1D. They identify both cis and trans pQTLs and attempt to put their findings into context with published GWAS of proteins in adults. Additional studies such as colocalization and Mendelian randomization have been performed in effects to explore whether the pQTLs may have an effect on T1D.

This is an interesting concept. However, I found the manuscript difficult to read, given the lack of headings of sections (abstract, intro, results, discussion). I also found the manuscript to be lacking detail, especially in the methods and results, therefore this manuscript needs more work to be publishable. There are some questions I have about some of the approaches used in the methods. I also think the interpretation of results may not be very accurate. Specific details are below.

We have extensively re-written and formatted the manuscript, which is now divided into sections with headings and sub-headings. We have also modified the text to include more information in all sections. We hope the new version of the manuscript is now easier to read.

Introduction

- Very brief introduction with 5 references. Authors need to give a more thorough overview of the current field.

Following the reformatting and extensive inclusion of detail, the introduction has also been extended.

Methods

Even though a reference to additional study information has been provided, factors such as the age of the children is important for the current study and therefore should be included in the manuscript. How do they define newborn?

For GPPAD participants included in this study, the median sampling age was 2 days and this information is now included in the manuscript.

If all are high risk of T1D, and no controls, will important signals for T1D be missed in the GWAS
The newborns enrolled in the GPPAD study have a >10% genetic risk of developing multiple beta-cell autoantibodies by the age of 6 years. The estimated prevalence of T1D in these high-risk infants was estimated to be 1.1%. Given the relatively low estimated prevalence, although much higher than the general population, we expect that T1D GWAS signals affecting the risk of developing the condition would remain detectable, albeit with likely reduced effect sizes.

Therefore, while it is possible that these signals did not reach genome-wide significance, we identified colocalization signals with T1D that were previously missed by Olink studies but identified in SomaScan pQTL studies, such as IL7R. On the other hand, pQTL signals specific for T1D would be more visible in this cohort. In support of this, we have added a new analysis in the revised manuscript that focuses on a subset of UKBB study participants with T1D, which allowed us to replicate three T1D-specific signals that were not significant in the full UKBB cohort.

Supp Table 2 lacks detail on each study and why they are being highlighted.

We have provided more information about the database in the manuscript and with reference to the original publication which contains the full description. We have also included more information about each study in Supplementary Table 2.

Line 222 - what does problematic proteins and problematic samples mean? What is this based on? Is it limit of detection?

We apologize for the vague description. We have now described in more detail the definitions for exclusion in the methods and have also added the number of excluded samples and proteins. Specifically, “problematic” was referring to the manufacturer’s assay and sample QC warnings.

Line 262, h_4 of > 0.6 defined as probable colocalization. This may be a bit too lenient.

The H_4 threshold for colocalization can vary across studies, with recent examples using thresholds as low as 0.5

Gudjonsson, A., Gudmundsdottir, V., Axelsson, G.T. et al. A genome-wide association study of serum proteins reveals shared loci with common diseases. *Nat Commun* 13, 480 (2022). <https://doi.org/10.1038/s41467-021-27850-z>

Oliva, M., Demanelis, K., Lu, Y. et al. DNA methylation QTL mapping across diverse human tissues provides molecular links between genetic variation and complex traits. *Nat Genet* 55, 112–122 (2023). <https://doi.org/10.1038/s41588-022-01248-z>

Further review of IL7R colocalization results shows that $H_4/(H_4 + H_3)$, which represents the probability of colocalization conditional on the presence of a causal variant, as described by

Zuber, V., Grinberg, N.F., Gill, D. et al. Combining evidence from Mendelian randomization and colocalization: Review and comparison of approaches. *The American Journal of Human Genetics*, 109, 5 (2022). <https://doi.org/10.1016/j.ajhg.2022.04.001>

is 0.68 and that $H_4/H_3 > 2$, indicating that the probability of the two traits sharing the same causal variant is twice as high as the probability of the two traits having two independent causal variants. However, we acknowledge that we cannot fully exclude the latter given that H_3 is non-zero ($H_3 = 0.16$). We believe these results are still noteworthy, and we have revised the manuscript to categorize the results as having high support ($H_4 > 0.8$) and medium support ($H_4 > 0.6$).

From line 267 - description of Mendelian randomization far too brief. Which GWAS were used for exposure and outcome? What p value for SNPs?

The methods section of the MR analysis has now been extended with substantially higher level of detail.

Did authors perform any power calculations for this GWAS? Small sample size used.

We agree with the reviewer that, compared to biobank-scale projects, the sample size is modest. However, it is larger or only moderately smaller than other recent pQTL studies such as

Gilly, A., Park, YC., Png, G. et al. Whole-genome sequencing analysis of the cardiometabolic proteome. *Nat Commun* 11, 6336 (2020). <https://doi.org/10.1038/s41467-020-20079-2> – sample size = 1328

Koprulu, M., Carrasco-Zanini, J., Wheeler, E. et al. Proteogenomic links to human metabolic diseases. *Nat Metab* 5, 516–528 (2023). <https://doi.org/10.1038/s42255-023-00753-7> – sample size = 1180

Steinberg, J., Southam, L., Roumeliotis, T.I. et al. A molecular quantitative trait locus map for osteoarthritis. *Nat Commun* 12, 1309 (2021). <https://doi.org/10.1038/s41467-021-21593-7> – sample size = 115

Among these, Koprulu *et al.* already demonstrated that smaller cohorts can still contribute to the identification of new biologically relevant proteogenomic links to disease. We hope that the reviewer will agree with us that this is the first proteogenomic study from newborn dried blood spots (median of 2 days) and, therefore, not comparable to previous studies on easily accessible tissues (blood) from adults.

We have also added a deep comparison of our results to the extended UKBB pQTL data and have also generated new data on a subset of UKBB participants with T1D. We have shown that one of our novel *cis*-pQTL signals is also nominally significant in the full UKBB cohort but did not reach genome-wide significance ($p\text{-value} = 6.5 \times 10^{-7}$) and that 3 signals are specific to the T1D-UKBB subgroup.

Therefore, despite the smaller sample size, our findings provide novel insights on the genetic regulation of protein levels in this valuable newborn cohort. To further address power concerns, we have also generated power curves and overlaid them to our results (supplementary figure 3) demonstrating that, given the observed effect sizes, we have sufficient power to detect the reported novel pQTLs.

Figure 1 - key for p value threshold unclear, all points for p value thresholds look the same. Points overlaying which changes the transparency of points. Maybe use size of the points for p value scale. The new figure included in the manuscript now uses the point size instead of transparency. We believe the suggestion improved the readability of the figure and we thank the reviewer for the suggestion.

When referencing Figure 1 - could make more specific references to Figure 1A/1B. Following the reformatting of the manuscript, Figure 1A has been moved to the supplementary materials and referencing Figure 1B is no longer needed.

Figure 2 - Axis labelled protein is confusing, how are these sorted?
The proteins used in Figure 1A are alphabetically sorted. This is now explained in the figure caption.

Authors could provide more of a summary of the pQTLs identified more broadly. We have included more general information about the identified pQTLs, such as the number of *cis*-pQTLs regulating protein levels in *trans* and distance of lead *cis*-pQTLs from their gene target transcription start site.

They mentioned that one SNP is *cis* for one protein and *trans* for another. What about all the other pQTLs from the GWAS? How many *cis* pQTLs were also *trans* pQTLs for other proteins?
As also suggested in the comment above, we have now added an extended summary of the pQTL results, which also includes the number of *cis* signals which were also *trans* for other proteins.

From line 118 - if the SNP for CTRB1 also associates with other proteins, does this suggest that there may be pleiotropy? Is the MR estimate truly estimating an effect of CTRB1 on T1D?
The regulatory mechanisms of *trans*-pQTL signals are not known. For eQTLs, which we might assume follow the same mechanisms, they are thought to be at least partially mediated through *cis* effects

Yang, F., Wang, J., The GTEx Consortium et al. Identifying *cis*-mediators for *trans*-eQTLs across many human tissues using genomic mediation analysis. *Genome Res.* 27, 1859-1871 (2017).
<https://doi.org/10.1101/gr.216754.116>

If that is the case for this locus, the *trans*-pQTL signals would be mediated through *CTRB1 cis*-pQTL. The observation that the *cis* and *trans* colocalizing signals share that same set of SNPs would indeed support this assumption but future experimental studies will be needed to confirm it.

Unclear where the discussion starts.

The whole manuscript has been re-formatted, and a clear Discussion section is now included.

Limitations need to be addressed.

The Discussion now includes a paragraph detailing the limitations of the study.

Sentence on 149 does not seem justified - you have identified pQTLs within a high-risk group, but have not shown that the levels of this protein are different compared to controls.

We agree with the reviewer and the sentence has now been modified:

“Whether the novel results that could not be identified by biobank-scale studies in adults, are due to increased risk of T1D in absence of the condition, or to newborn age, or a combination of the two, will require future replication.”

No conflict of interest or author contributions

We have now added the conflict of interest and author contributions statements to the manuscript.

Reviewer 3

The authors construct a genome-wide pQTL map of 1985 proteins in 695 newborns at high risk of T1D, showing 535pQTLs (66%*cis*) in 437 unique proteins of which 62 are novel. Five pQTL, selected from overlap with T1D associated variants, show colocalisation with T1D susceptibility loci with Mendelian Randomisation implicating them in the aetiology of T1D.

The novel variants identified are difficult to interpret given the technical concerns over the platform (affinity impacted by protein-coding variants) and a lack of validation. Most pQTL identified are previously reported and two of the three 'causal' variants identified (*IL7R*, *CTRB1*) have supportive evidence on another platform as pQTL. The principle novelty in what is shown is their colocalisation with T1D risk variants and the MR analysis indicating a causal role in T1D. Other pQTL are effectively validation of reported associations in newborns or unvalidated new pQTL in a distinct and much smaller cohort. There is clear discussion of previous functional work on the 2 causal variants but no new data presented. The authors claim that their results may allow inclusion of protein quantification as biomarkers for risk stratification or for drug target identification but it isn't clear what this might add and no new data.

Methods

Use an established screening platform (GPPAD) to identify high risk children and an ongoing therapeutic trial of oral insulin nested within that platform.

Use dried blood spot from newborns to quantify 1985 proteins using OLink platform.

pQTL analysis on those proteins with colocalization and causal inference (mendelian randomisation) to investigate link to T1D. No discussion of why restricted to MR on *cis* only variants with *trans* variants excluded.

We thank the reviewer for their comment. Both *cis* and *trans* signals are now reported for both coloc and MR analysis.

Not clear how multiple correction was handled during the identification of 'significant' coloc for the 5 variants (and not explicit over how many variants tested from the meta-analysis).

Colocalization analysis was done using the R package Coloc, a well-established Bayesian method that uses the full summary statistics from the pQTL and GWAS studies, centered around the GWAS lead variant +/- 1Mb, to calculate the poster probability of the two signals sharing the same causal variant in the locus. This is also now stated in the main text. The strength of the method is that it evaluates all SNPs within the locus, rather than a single SNP, accounting for the linkage disequilibrium (LD) structure among SNPs. The number of SNPs used for each colocalization test is reported in the supplementary tables 5 and 6 (column "nvariants") for *cis* and *trans* signals, respectively.

Correction of the multiple protein-phenotype colocalizations is a known analytical challenge with the analysis. Colocalization tests return posterior probabilities (PP) for different scenarios (e.g., PP for the presence of a shared causal variant, presence of separate causal variants, etc.) and it is not straightforward how this information could be incorporated in the prior probabilities of the Bayesian tests.

Recent large consortia studies, such as GTEx and Open Targets, have also not applied multiple-testing correction for all QTL-GWAS locus pairs in colocalization analyses.

For example, GTEx Consortium, did not perform multiple-testing correction in their colocalization study between eQTL data from 49 tissues and 87 GWAS. Similarly, Open Targets did not apply correction in their colocalization analyses between multiple QTL datasets and over 1,000 GWAS studies.

The GTEx Consortium, The GTEx Consortium atlas of genetic regulatory effects across human tissues. *Science* 369, 1318-1330(2020). <https://doi.org/10.1126/science.aaz1776>

Mountjoy, E., Schmidt, E.M., Carmona, M. et al. An open approach to systematically prioritize causal variants and genes at all published human GWAS trait-associated loci. *Nat Genet* 53, 1527–1533 (2021). <https://doi.org/10.1038/s41588-021-00945-5>

Results

Fig 1A is not a very helpful way to present the findings as much of what is shown is not clearly interpretable. Little actual data is presented - for example we are not shown the proteins tested, which overlap with the adult cohort and which are 'new'.

We have removed Figure 1A from the main manuscript since, in agreement with the reviewer, we believe the figure does not provide enough information to be included in the main text. We believe the figure still provides an interesting genome-wide overview of the results that it is not otherwise possible in 2D and we have therefore moved it to a new supplementary materials file. We have improved Figure 1B, which is now Figure 1, to show which results are novel and which are known. This information is also available in Supplementary Figure 3.

The colour legend for Fig1B appears to relate to cis/trans category only and not the magnitude of significance as presented.

The figure has been modified to clearly display the magnitude of effect. It now also displays novelty, as described in the previous comment.

It would also be relevant to show which pQTL that are reported elsewhere were not detected amongst the 467 proteins with a pQTL in newborns. It is claimed that pQTL are largely stable between newborns and adults, although we don't see what % of the 1985-467 = 1518 proteins that don't show a pQTL here, did show one in adults (I accept the power of detection is very different, but that is a relevant limitation to the discovery here).

The UKBB study had a much larger sample size and identified 2627 pQTL-targeted assays (as reported in Table 1 of Eldjarn et al.), which corresponds to 90% of the proteins analyzed

Eldjarn, G.H., Ferkingstad, E., Lund, S.H. et al. Large-scale plasma proteomics comparisons through genetics and disease associations. *Nature* 622, 348–358 (2023).
<https://doi.org/10.1038/s41586-023-06563-x>

We have now added an in-depth comparison to both the publicly available UKBB pQTL summary statistics and to a new analysis undertaken for the purposes of the revised manuscript, on a subset of UKBB participants with type 1 diabetes. We show that there is strong correlation ($R=0.77$; Supplementary Figure 4) between the effect sizes from our pQTL analysis and the full UKBB dataset. We also show replication of specific pQTLs in individuals with type 1 diabetes. We agree with the reviewer that the power difference between this study and the UKBB is substantial. This is reflected in the fact that 1329 (87.8%) of proteins that did not show a pQTL here, did show one in adults.

This power difference remains a major obstacle for the study's design and is reflected in this result. This has large implications for interpretation of the results presented. Underpowered analysis can generate false positives as well as miss true positives. While the concordant direction of effect in the larger cohort is supportive, it isn't validation (I presume these are not similarly significant in the other cohort otherwise this would be stated).

We aren't told how the 1985 proteins are selected (presumably based on availability through Olink panels, but those are more extensive than the proteins tested here). The 1985 protein assays, out of 2941 initial assays included in the Olink panels, were selected based on passing stringent QC. The total number of measured proteins and the QC steps are now fully detailed in the methods.

The link between variants and detection is clearly an issue for all new and potentially for previously-reported pQTL also. Reader cannot easily define what is valid given the data presented. The issue potentially relates to any O-link quantified protein that is not validated with a secondary method, and that appears to include most of what is presented here.

We agree that affinity-based assay measurements could be affected by protein-structure-altering variants which would result in a *cis*-pQTL signal. For the proteins measured by both technologies, we have therefore now added information regarding correlation of protein level measurements between Olink and SomaScan from publicly available datasets. We have also queried eQTL databases to show that the *cis* variants are not only associated with protein levels but also with gene expression. For the majority of cases, we found good agreement between Olink and SomaScan measurements and all 4 novel *cis*-pQTLs have been detected as eQTLs in blood with the same direction of effect. While these additional analyses show that the reported results are robust, we agree that larger proteogenomic studies on other technologies such as mass-spec will be needed in the future for further validation.

This is a major caveat and limits confidence in any result not validated with a second method. This affects much of what is shown. I don't see why this is needed in the future, and not for the work presented here?

Minimal discussion of the 'New' pQTL and why they are detected here and not elsewhere?

We have now performed a KEGG pathways over-representation analysis of the proteins targeted by novel pQTLs, and found a significant enrichment for the insulin signaling pathway. In the absence of a comparable newborn cohort, we have analyzed UKBB genotype and Olink data for a subset of participants with type 1 diabetes ($N=61$ adults). Although none of the novel *cis*-pQTLs reached significance in this smaller subgroup, we successfully replicated 3 *trans* signals related to T1D biology. SPTLC, a protein involved in the sphingolipid metabolism whose *cis*-pQTL was previously identified as associated with T-helper proportions in a T1D cohort

I accept the challenges of obtaining a replication cohort here, but the risk of false discovery is high and the above is a failure of validation. The majority of observed associations don't validate in the larger public cohort either, highlighting the risk of false positive associations (or, you could argue, associations that are unique to neonates vs adults).

Chu X., Janssen WM. A., Koenen H., et al. A genome-wide functional genomics approach uncovers genetic determinants of immune phenotypes in type 1 diabetes eLife 11:e73709 (2022) <https://doi.org/10.7554/eLife.73709>

The IQ Motif Containing GTPase Activating Protein 2 (IQGAP2) was instead targeted by a SNP that sits 300kb from the insulin gene promoter. All of these new analyses and results are presented and discussed in the revised manuscript. the WAP, Kazal, immunoglobulin, Kunitz, and NTR domain-containing protein 2 (WFIKKN2), a loss-of-function variant affecting WFIKKN2 circulating protein levels has been associated with HOMA-IR levels

Ngo D, Benson MD, Long JZ, et al. Proteomic profiling reveals biomarkers and pathways in type 2 diabetes risk. JCI Insight. (2021) Mar 8;6(5):e144392 <https://doi.org/10.1172/jci.insight.144392>

No description of possible mechanism, although not likely to be simple given expression of ICA1 in pancreas and in Treg (with expression here presumably dictated more by the latter). ICA1 has been previously associated with T1D but its role in the disease is uncertain. As pointed out by the reviewer, it is expressed in both pancreas and Treg

Pesenacker A.M., Chen V., Gillies J. et al. Treg gene signatures predict and measure type 1 diabetes trajectory. JCI Insight (2019);4(6):e123879. <https://doi.org/10.1172/jci.insight.123879>

Karges W., Pietropaolo M., Ackerley C.A. et al. Gene Expression of Islet Cell Antigen p69 in Human, Mouse, and Rat. Diabetes 1 April 1996; 45 (4): 513–521. <https://doi.org/10.2337/diab.45.4.513>

both of which have a role in T1D. It is also observed in testis and there is a possibility that it has a neurotransmitter activity. Reference to its expression in the pancreas and its T1D-associated expression in Tregs has now been added to the discussion, but we have chosen not to speculate on mechanism.

Validation of most pQTL is not necessary (as they are previously reported and hence this study is validation, only in newborns). However, it would be necessary to validate the 'new' findings, both technically (with another platform) and in another cohort.

We thank the reviewer for this comment. As mentioned in the response to the comment further above, we have attempted replication of novel signals by accessing the full UKBB summary statistics to see if the novel results were also nominally significant in the UKBB. We have also extracted publicly available data on the correlation of Olink and SomaScan protein measurements and eQTL data from GTEx and eQTLGen to exclude that the novel results are technical and not biological. Finally, we gained access to UKBB genotype and Olink measurements in the subset of participants with type 1 diabetes. Altogether, we could replicate a large part of the findings and identified type 1 diabetes specific pQTLs. For those signals we could not replicate, new data for a proteogenomic study on a similar cohort, but with a different technology, will have to be generated. Given the scarcity of materials available from dried blood spots from newborns, such data generation on the same newborns is currently not possible.

“Altogether, we could replicate a large part of the findings and identified type 1 diabetes specific pQTLs”

I don't think this statement is supported by what is presented. The key new pQTL didn't validate with concordant effect direction being a much weaker signal.

Only 2 of the three cis pQTL run validated in Somalogic data and no attempt to validate the association with a secondary technology.

Olink and SomaScan are now the leading platforms for proteogenomic studies. Very scarce blood pQTL data are available from complementary technologies, such as mass-spec or ELISA. Our definition of novelty is based on comparison to an extensive database of 46 studies, 6 of which are from mass-spec measurements. Replication of the results using publicly available data is therefore not possible. We have now further attempted replication by accessing the full summary UKBB pQTL summary statistics and by generating new UKBB pQTL data on a subset of UKBB participants with T1D. Given the scarcity of materials available from dried blood spots from newborns, and the complexity of optimizing a new assay for dried blood spots, it was not possible to further replicate the results in newborns.

This statement is a more accurate caveat to the data presented, although unavoidably limits the interest and confidence in the findings.

Trans-pQTL not run with mendelian randomisation and not made clear why the restriction to cis-variants was made.

Both cis and trans signals have now been included in the colocalization and Mendelian randomization analysis.

CTRB1 variant has several previously-described associations including pancreatic volume and T1D. The principal novel result here is the coloc and MR analysis suggesting causality. No new data on the putative mechanism is presented.

We agree that previous studies have found a correlation between the levels of the colocalizing proteins and T1D. However, these studies did not account for genetic factors, leaving the causal relationship between the protein levels and disease state unclear. Our findings demonstrate that genetic variants are linked to both protein levels and disease well before the onset of symptoms, at birth. This suggests that the direction of effect is SNP → protein → T1D, since a reverse temporal association is not possible. These results provide unique information that further strengthen previous links between these proteins and T1D. We agree that understanding the functional mechanism of the identified association is important but it is beyond the scope of the current study. We are currently planning to study the effect of CTRB1, as well as, IL7R and APOBR in a longitudinal setting once the clinical trial has ended.

Thanks – this is a fair response and a good point.

IL7R - previous functional characterisation of the variant but nothing new presented here.

Please see response to the comment above.

Central to the discussion is the proposal that new pQTL signals present at birth but not later could indicate an early pathology. This is very speculative, particularly as there is no validation of the 'new' pQTL signals, with no colocalisation or MR analysis of them.

With the new analyses, we have now shown that the new pQTLs are enriched for the insulin signaling pathway. We have also identified 3 novel pQTLs as possibly T1D specific, with replication in a small

subset of UKBB, one of which has previously been reported as a T1D-specific QTL. The discussion has been extensively re-written and we hope the reviewer will agree with our interpretation of the results.

Similarly, it is claimed (in both abstract and discussion) that the proteins could be used as supportive biomarkers for early diagnosis, supporting GRS. However, GRS already have access to the SNP variants associated with T1D on which these pQTL are based and no data is presented to support their utility as biomarkers or therapeutic targets (although they could have been considered as either before this study given the previously demonstrated SNP association and, in some cases, demonstrated protein level association with disease).

We have removed reference to the use of the proteins as supportive biomarkers from the abstract since we have no data. However, we feel that it is warranted to briefly speculate that these have potential as predictive biomarkers with or without a polygenic risk score in the discussion. In combination with a PRS, there may be added value since although there is an association with the predisposing SNP, the values within genotypes of the SNP vary. As indicated, we do not have the data set to test whether the protein concentration is predictive for islet autoantibodies or T1D within genotypes. Furthermore, pQTLs only explain part of the protein heritability. Including the dynamic protein levels resulting from gene-environment interactions may complement the fixed genetic risk of the PRS model. To support this, there are several publications on prediction models based on UKBB Olink data showing increased accuracy of including protein levels for disease prediction

Carrasco-Zanini, J., Pietzner, M., Davitte, J. et al. Proteomic signatures improve risk prediction for common and rare diseases. *Nat Med* 30, 2489–2498 (2024).
<https://doi.org/10.1038/s41591-024-03142-z>

It has also been shown that protein-based prediction models can be more accurate than those based on PRS, but it is currently not known whether genetic-agnostic protein selection is better than genetically informed selection, or whether the two overlap. Whether the inclusion of the identified pQTLs or the protein levels in the GPPAD PRS model could help increase its accuracy will need to be addressed but it is beyond the scope of this manuscript.